

# Estimating snow depth over Arctic sea ice from calibrated dual-frequency radar freeboards

Isobel Lawrence[1], Michel Tsamados[1], Julienne Stroeve[1,2], Thomas Armitage[3], and Andy Ridout[1]

[1]Centre for Polar Observation and Modelling, Earth Sciences, University College London, London, UK
[2]National Snow and Ice Data Center, University of Colorado, Boulder, CO, USA
[3]Jet Propulsion Laboratory, California Institute of Technology, Pasadena, California, USA

*Correspondence to:* Isobel Lawrence (isobel.lawrence.15@ucl.ac.uk)

**Abstract.**

Snow depth on sea ice remains one of the largest uncertainties in sea ice thickness retrievals from satellite altimetry. Here we outline an approach for deriving snow thickness that can be applied to any coincident freeboard measurements after calibration with independent observations of snow and ice freeboard. Freeboard estimates from CryoSat-2 (Ku-band) and AltiKa (Ka-band) are calibrated against data from NASA's Operation IceBridge (OIB) to align AltiKa to the snow surface and CryoSat-2 to the ice/snow interface. Snow depth is found as the difference between the two calibrated freeboards, with a correction added for the slower speed of light propagation through snow. We perform an initial evaluation of our derived snow depth product against OIB snow depth data by excluding successive years of OIB data from the analysis. We find a root-mean-square deviation of 4.9, 6.5, 6.7 and 7.6 cm between our snow thickness product and OIB data from the springs of 2013, 2014, 2015 and 2016 respectively. We further demonstrate the applicability of the method to ICESat and Envisat, offering promising potential for the application to CryoSat-2 and ICESat-2, when ICESat-2 is launched in 2018.

## 1 Introduction

The addition of snow on sea ice, given its optical and thermal properties, generates several effects on the climate of the polar regions. Owing to its large air content, snow has a thermal conductivity ten times less than that of ice (Maykut and Untersteiner, 1971). During the winter freeze-up, it forms an insulating layer that reduces heat flow from the ocean to the atmosphere and slows the rate at which seawater freezes to the bottom of the ice, dampening further ice growth (Sturm et al., 2002).

Snow has an optical albedo in the range of 0.7-0.85, compared to 0.6-0.65 for melting white ice (Grenfell et al., 1977). At the onset of the melt season, short-wave solar radiation is reflected from the surface, limiting ice melt. These properties make snow on sea ice important in energy budget considerations and Arctic snow depth estimates could be usefully assimilated to improve weather and sea ice forecasting.

As well as its climatic importance, snow depth plays a key role in the retrieval of sea ice thickness from satellite altimetry. Over the past two decades both radar (e.g. ERS-2, Envisat, CryoSat-2) and laser (e.g. ICESat) altimeters have enabled sea ice thickness to be measured from space (Laxon et al., 2003, 2013; Kwok and Cunningham, 2008). The implications of uncertain snow depth is different for each measurement approach. For the radar case, snow depth plays a two-stage role: First, under





the assumption that the radar pulse penetrates through the snow to the ice/snow interface (Beaven et al., 1995), a correction to account for the slower speed of light propagation through the snow pack is required in order to convert radar freeboard to ice freeboard. This correction is given by:

$$f_i = f_r + h_s\left(\frac{c}{c_s} - 1\right)$$ (1)

where $f_i$ is the ice freeboard, $f_r$ is the radar freeboard measured by the radar altimeter, $h_s$ is the snow depth and $c$ and $c_s$ are the speed of light in a vacuum and in snow, respectively.

Secondly, the added weight of a snow cover alters the buoyancy of the sea ice floe, therefore snow thickness is required to convert sea ice freeboard to thickness $t_i$. Assuming hydrostatic equilibrium, sea ice thickness is given by:

$$t_i = \frac{\rho_s}{\rho_w - \rho_i}h_s + \frac{\rho_w}{\rho_w - \rho_i}f_i$$ (2)

where $\rho_s$, $\rho_w$, and $\rho_i$ are the densities of snow, water and ice respectively.

For laser altimetry [e.g. Kwok et al. (2004, 2007)], where it is assumed that the laser does not penetrate the snow and thus the return echo comes from the air/snow interface (Zwally et al., 2002), the freeboard $f_l$ represents the height of combined ice plus snow layers above sea level and the hydrostatic equation becomes:

$$t_i = \frac{\rho_s - \rho_w}{\rho_w - \rho_i}h_s + \frac{\rho_w}{\rho_w - \rho_i}f_l$$ (3)

Giles et al. (2007) used typical values and their uncertainties for quantities in Eqs. (2) and (3) to perform an error sensitivity analysis on retrieved sea ice thickness. They found that snow depth uncertainty represents the dominant error contribution for both the radar and laser case; 48% and 88% of the total error respectively. Their typical snow depth uncertainty of 0.11 m contributed a 0.32 m error on sea ice thickness from radar altimetry, and 0.7 m error on sea ice thickness from laser altimetry.

Only satellite-derived snow depth estimates can offer the spatio-temporal resolution required for sea ice thickness derivation,
but retrieving snow depth from space has proven challenging and is an ongoing effort for the sea ice community. This paper addresses this critical data gap by demonstrating an approach for deriving snow thickness that can be applied to any coincident freeboard measurements after calibration with an independent observation of snow and ice freeboard. Before outlining the proposed method and introducing our Dual-altimeter Snow Thickness (DuST) product, we first review the most successful existing approaches for retrieving snow depth from satellites, and discuss their limitations.

## 2    Existing satellite snow depth products

Existing methods to retrieve snow depth from satellites have historically relied on using relationships between passive microwave brightness temperatures and snow thickness. The granular nature of snow acts to scatter and dissipate microwave energy radiating from the Earth's surface, reducing the surface brightness temperature. Markus and Cavalieri (1998) developed the first snow-depth-on-sea ice algorithm on the basis of two features of this snow scattering: 1) The linear reduction in bright-
ness temperatures with increasing snow depth for a given frequency, and 2) higher attenuation at higher frequencies. Using data




over Antarctic sea ice from the Defense Meteorological Satellite Program (DMSP) special sensor microwave/imager (SSM/I), they compared the spectral gradient ratio of the 19 and 37 GHz vertical polarization channels with in-situ snow depth data in order to express snow depth as a function of brightness temperature. The algorithm was later developed for application to Arctic sea ice using data from the Advanced Microwave Scanning Radiometer-EOS (AMSR-E), but due to the inability to dis-
tinguish signatures from snow and multi-year ice, the available AMSR-E data product is limited to seasonal ice only (Comiso et al., 2003; Markus and Cavalieri, 2012). Furthermore, subsequent studies have demonstrated the sensitivity of the retrieved snow depth to snowpack conditions and surface roughness (Stroeve et al., 2005; Powell et al., 2006).

In another study using passive microwave, Maaß et al. (2013) utilised a frequency of 1.4 GHz (L-band), measured by the European Space Agency's Soil Moisture and Ocean Salinity (SMOS) satellite. Although snow is transparent to L-band
frequencies, i.e. the large wavelengths are not attenuated by the snow, their model-based study found brightness temperatures from the ice increased at L-band frequencies when a snow layer was present due to its insulating properties and the dependence of ice emissivity on temperature.

Using a radiative transfer model, they tested the impact of 0-70 cm varying snow thickness on L-band brightness temperatures for a number of scenarios (in which ice temperature, thickness, salinity, and snow density varied within a realistic range).
The snow depth which produced a brightness temperature most comparable (smallest root mean square deviation and best correlation coefficient) to SMOS brightness temperature was then compared with snow thickness from Operation IceBridge in order to asses which scenario performed best. Snow depths produced by this scenario correlated well (root-mean-square deviation = 5.5 cm) up to model-generated depths of 35 cm, but overestimated snow depth thereafter, owing to the desensitisation of brightness temperatures when snow depth increases above 35 cm. Furthermore, this approach requires that the values for
the input parameters (ice temperature, thickness, salinity, and snow density) are assumed valid everywhere. In reality, these parameters vary in space and time and the authors express the need to develop the methodology further to allow regional and temporal variability of model input parameters. At time of publication of this study, no SMOS snow depth product has been made publicly available.

A new approach to snow depth retrieval from satellites was offered by Guerreiro et al. (2016), who demonstrated the potential
to estimate snow depth by comparing retrievals from coincident satellite radar altimeters operating at different frequencies. Snow depth over Arctic sea ice was retrieved by differencing elevation retrievals from AltiKa (Ka-band radar satellite altimeter, 2013-present) and CryoSat-2 (CS-2) (Ku-band radar satellite altimeter, 2010-present). To investigate the penetration properties of the two radar altimeters, the authors simulated penetration depth as a function of snow grain size, under different temperature and density conditions, derived from the equation for the extinction coefficient of the radar signal. Based on these model
simulations and using existing field campaign data to characterise average snow grain sizes, AltiKa was designated with a maximum penetration depth of 0, i.e. no penetration, and CS-2 a maximum penetration of 1, i.e. full snow penetration, across the Arctic basin.

In contrast to this, Armitage and Ridout (2015) derived AltiKa freeboard for the first time and investigated the spatial variability of AltiKa and CS-2 penetration. By comparing the freeboards retrieved from each satellite with an independent
measurement of ice freeboard from NASA's Operation IceBridge (OIB), radar penetration at a local grid-scale level was quan-



tified. Under the assumption that multi-year ice and first-year ice characterise snow and ice packs with distinctive penetrative properties, an average value for radar penetration factor was found for each satellite over each ice type. Though limited to the spring due to the availability of OIB data and therefore not necessarily representative of penetration properties throughout the year, the study highlights the importance of accounting for regional differences in penetration depth.

Guerreiro et al. (2017) compared freeboards from Envisat, a Ku-band pulse-limited altimeter, with those from the CS-2 SAR system. Since both altimeters operate at the same frequency, they are expected to penetrate to the same depth and therefore retrieve comparable freeboards. The study found Envisat was biased low compared with CS-2, attributed to differences in footprint size and the effect of the retracker on SAR and pulse-limited waveforms (discussed in Sect. 3.3). Schwegmann et al. (2016) performed a similar Envisat / CS-2 freeboard comparison over Antarctic sea ice and similarly found a bias on Envisat's
freeboard attributed to its larger footprint.

These results suggest that the freeboard difference between AltiKa and CS-2 in Armitage and Ridout (2015) may not be a result of penetration differences alone, but subject to biases due to differences in sampling area and processing techniques. AltiKa has a smaller footprint than that of Envisat (1.4 km compared with 2-10 km); nevertheless we would expect the impact of its different footprint with respect to CS-2 to introduce a bias like that seen in the Envisat data.

Building on the methodology of Armitage and Ridout (2015), we make use of independent snow depth and laser freeboard data from OIB to asses the deviation of AltiKa and CS-2 satellite freeboards from the snow surface and snow/ice interface respectively. We assume this deviation to result from the combination of competing effects; penetration depth, biases due to sampling area and surface roughness, and effect of the threshold retracker on the satellite waveforms. Following Guerreiro et al. (2017), we compare each satellite's deviation from its expected dominant scattering horizon ($\Delta f$) against satellite pulse
peakiness. Using the relationships between $\Delta f$ and pulse peakiness, we calibrate both AltiKa and CS-2 freeboards to bring them in line with the snow surface and snow/ice interface respectively. We then estimate snow depth as the difference between the calibrated AltiKa and CS-2 freeboard. The advantage of our approach is its applicability to any freeboard data sets providing they can be calibrated with an independent measure of snow/ice freeboard.

## 3   Data and methods

### 3.1   AltiKa

The Satellite for Argos and AltiKa (herein referred to as AltiKa), was launched in spring 2013 as a joint mission between the Centre National d'Etudes Spatiales (CNES) and the Indian Space Research Organisation (ISRO). AltiKa's pulse-limited Ka-band radar altimeter, which operates at a central frequency of 35.75 GHz, retrieves surface elevations up to 81.5°. The first sea ice freeboard estimates using AltiKa data were presented in Armitage and Ridout (2015), who used a 'Gaussian plus
exponential' retracker to retrieve lead elevations (after Giles et al. (2007)) and a 50% threshold retracker over floes. AltiKa freeboard data used in this study are derived using the same processing algorithm and the reader is referred to the supplementary material in Armitage and Ridout (2015) for further details.



## 3.2 CryoSat-2

CS-2 was launched by the European Space Agency in 2010, tasked with the specific role of monitoring the Earth's cryosphere. The satellite has an orbital inclination of 88°, giving it far better coverage over the poles than previous radar altimeters, and, unlike AltiKa, CS-2 employs along-track SAR processing to achieve an along-track resolution of approximately 300 m, improving the sampling of smaller floes and making it less susceptible to snagging from off-nadir leads (Wingham et al., 2006). As with AltiKa, lead elevations are retrieved using the 'Gaussian plus exponential' model fit and for floes a 70% threshold retracker was determined as offering the best average elevation from CS-2's unique SAR waveforms (Tilling et al., 2017). The CS-2 freeboard data used in this study were processed by the Centre for Polar Observation and Modelling (CPOM) and readers are referred to Tilling et al. (2017) for further details on the method.

## 3.3 Sources of AltiKa/CryoSat-2 freeboard bias

We define AltiKa/CS-2 freeboard bias as the portion of the AltiKa minus CS-2 freeboard difference that does not originate from the difference in snow penetration of the two radars. In line with radar theory (Rapley et al., 1983) and in light of recent findings by Guerreiro et al. (2017) we expect such a bias to be the result of the difference in footprint sizes between the two altimeters and the consequences of this during freeboard processing. The differences between AltiKa and CS-2 of interest to this study are summarised in Table 1.

**Table 1.** AltiKa and CS-2 (SAR mode) operation characteristics

|  | Period of operation | Operating frequency | Footprint size | Footprint area | Sampling interval | Latitude limit |
|---|---|---|---|---|---|---|
| AltiKa | Feb 2013 - present | 35.75 GHz (Ka-band radar) | 1.4 km diameter (pulse-limited) | 1.5 km$^2$ | 0.17 km | 81.5° |
| CryoSat-2 SAR | April 2010 - present | 13.57 GHz (Ku-band radar) | 0.3/1.7 km along/across track (Doppler cell) | 0.5 km$^2$ | 0.3 km | 88° |

In an initial stage of AltiKa and CS-2 freeboard processing, waveforms are classified as either lead or floe according to thresholds for Pulse Peakiness (hereafter PP), defined as:

$$\text{PP} = N \frac{p_{max}}{\Sigma_i \, p_i}$$

where $N$ is the number of range bins above the 'noise floor' (calculated as the mean power in range bins 10-20), $p_{max}$ is the maximum waveform power (the 'highest peak'), and $\Sigma_i \, p_i$ is the sum of the power in all range bins above the noise floor (Peacock and Laxon, 2004). Waveforms originating from smooth, specular leads demonstrate a rapid rise in power followed by a sharp drop off, giving them a high PP. Returns from floes typically demonstrate a more gradual rise in power and slower drop-off, equivalent to a lower PP. PP can therefore be used to distinguish floe and lead returns, and eliminate those not clearly





identifiable as one or the other. For AltiKa(CS-2), waveforms with PP less than 5(9) are designated as originating from ice floes. Waveforms with PP greater than 18 are classified as leads for both satellites (Armitage and Ridout, 2015; Tilling et al., 2017).

Waveforms that exhibit a mixture of scattering behaviour will have a PP in the 'ambiguous' range (5<PP<18 for AltiKa and 9<PP<18 for CS-2) and are discarded. Since AltiKa has a larger footprint, its waveforms are more likely to be ambiguous and therefore discarded than CS-2, which can resolve smaller floes within the same region. The result of this is a bias in AltiKa towards higher freeboards (only larger floes, which tend to be thicker, are captured), especially over seasonal, lead-dense areas.

The impact of surface roughness on pulse-limited altimetry is well documented (e.g. Rapley et al. (1983); Raney (1995); Chelton et al. (2001)). Generally, a rougher surface leads to dilation of the footprint and a widening of the leading edge of the waveform return. For a homogeneously rough surface with a Gaussian surface elevation distribution, the 50% power threshold represents the mean surface elevation within the pulse-limited footprint. However, for a heterogeneously rough surface, such as that of multi-year sea ice, the waveform leading edge can take a complex shape, where the half-power point does not necessarily represent the average elevation within the footprint and using a 50% threshold retracker might lead to a biased surface height retrieval. Since AltiKa does not benefit from the along-track Doppler processing and effective sharpening of the waveform response that CS-2 does, it is more susceptible to a freeboard bias over rough sea ice due to this effect.

AltiKa is also more sensitive to off-nadir ranging to leads due to its larger footprint. Error occurs when off-nadir leads dominate the waveform, resulting in an overestimate of the range to the lead, an underestimate of sea surface height, and a positive bias on the local floe freeboard (Armitage and Davidson, 2014). To minimise this effect, lead waveforms for AltiKa are discarded if their backscatter per unit area, $\sigma^0$, is less than 24 dB, under the assumption that off-nadir leads return less power to the antenna compared with those at nadir (Armitage and Ridout, 2015). However, It is unlikely that this criteria eradicates the problem altogether and we expect that the freeboard bias due to snagging is larger in the AltiKa data compared to CS-2.

To overcome these problems, Guerreiro et al. (2016) employed degraded SAR mode CS-2 data in their comparison, where the synthetic Doppler beams are not aligned in time and are summed incoherently to obtain a pseudo-pulse-limited echo. Since this offers a footprint and waveform more closely resembling that of AltiKa, it was assumed that observed elevation differences between AltiKa and degraded CS-2 were the result of differences in snow penetration only.

Rather than separating the contributions of freeboard difference in this way, we instead adopt an approach that calibrates AltiKa freeboard to the level of the snow and CS-2 to the ice/snow interface. As such, penetration properties and sources of freeboard bias are corrected in one step without needing to consider the contribution of each. It is apparent that different freeboard products derived through different processing chains via different groups, are not consistent (Stroeve et al., 2018). Short of any evidence in support of which product is better, the appeal of this methodology is its applicability to any freeboard data sets. By calibrating satellite freeboards with an independent data set, biases are systematically corrected for.

### 3.4 Operation IceBridge

In order to evaluate the deviation of each satellite's retrieved elevation from its expected dominant scattering horizon (the snow surface for AltiKa and the snow/ice interface for CS-2), we use data from NASA's 2013-2016 OIB spring campaigns. At time





of publication, OIB products from 2014 to 2016 are only available in Quick Look format: a first-release, expedited version, which demonstrates reduced accuracy compared with the final release products (Kurtz, 2014). In the interest of consistency we also use the Quick Look product for 2013, and acknowledge that our calibrations will be improved with the release of archival OIB products.

Publicly available OIB products do not include a sea ice freeboard parameter but do output, at 40 m along-track resolution, coincident measurements of snow freeboard (i.e. total height of ice plus snow above water) from an ATM laser altimeter, and snow depth, retrieved with the Kansas Snow Radar to within 5 cm accuracy (Kurtz, 2014). Sea ice freeboard $f_i$ is retrieved by subtracting snow depth $h_s$ from the laser freeboard $f_l$. Re-arranging Eq. (1), ice freeboard is then converted to radar freeboard $f_r$ by:

$$f_r = f_i - h_s\left(\frac{c}{c_s} - 1\right) \tag{4}$$

The OIB radar freeboard represents the freeboard that would be retrieved by a satellite altimeter whose pulse penetrated through to the ice/snow interface (Armitage and Ridout, 2015). We choose a value of $c/c_s$ of 1.28 after Kwok (2014). In the following discussion, AltiKa and CS-2 freeboard refers to the *radar* freeboard, that is the freeboard retrieved before the correction for light propagation through the snow pack, given by Eq. (1), is applied.

## 3.5 AltiKa calibration with Operation IceBridge

For each day of the three spring campaigns 2013-2015, OIB laser freeboard data are averaged onto a 2°longitude x 0.5°latitude grid. Grid cells containing less than 50 individual points are discarded to remove speckle noise. Along-track AltiKa freeboard and PP data for the ±10 days surrounding the campaign day are then averaged onto the same grid and grid cells with less than 50 points are similarly discarded. This grid and time window were chosen because they offered the best spatial and temporal

resolution possible whilst ensuring enough coverage to minimise the noise.

Satellite freeboard and PP grids are then interpolated at the average position of the OIB data within each valid OIB grid cell. Further, high resolution (10 km gridded) ice type data from the Ocean and Sea Ice Satellite Application Facilities (OSI SAF, http://osisaf.met.no) are interpolated at the same point to determine whether multi-year or seasonal ice is being sampled. $\Delta f_{AK}$, defined as ATM laser freeboard minus AltiKa freeboard, plotted against AltiKa PP is shown in Fig. 1. Data from 2013,

2014 and 2015 and their corresponding linear regression fits are plotted in red, blue and grey respectively to demonstrate year to year consistency. Further, multi-year and first-year ice are distinguished by star and square markers in order to illustrate the variation of pulse peakiness, and thus roughness, with ice type.

The combined (all years) linear regression fit (CLRF) is shown by the black line and has slope of -0.16 and intercept of 0.76. The shaded area shows the 68% prediction interval about the CLRF, corresponding to a standard error (SE) on $\Delta f_{AK}$ of 9.4

cm. The CLRF is greater than zero for most PPs, implying that the freeboard needs to be increased to align with the snow/air interface, though moreso (∼0.2 m) for low peakiness values (rougher ice) than for high peakiness values (smoother ice), where the correction approaches zero. This suggests that freeboard over rough ice is biased low, which could be attributed to difficulty in identifying the average footprint surface elevation as outlined in Sect. 3.3. It could also suggest that AltiKa penetrates further

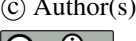


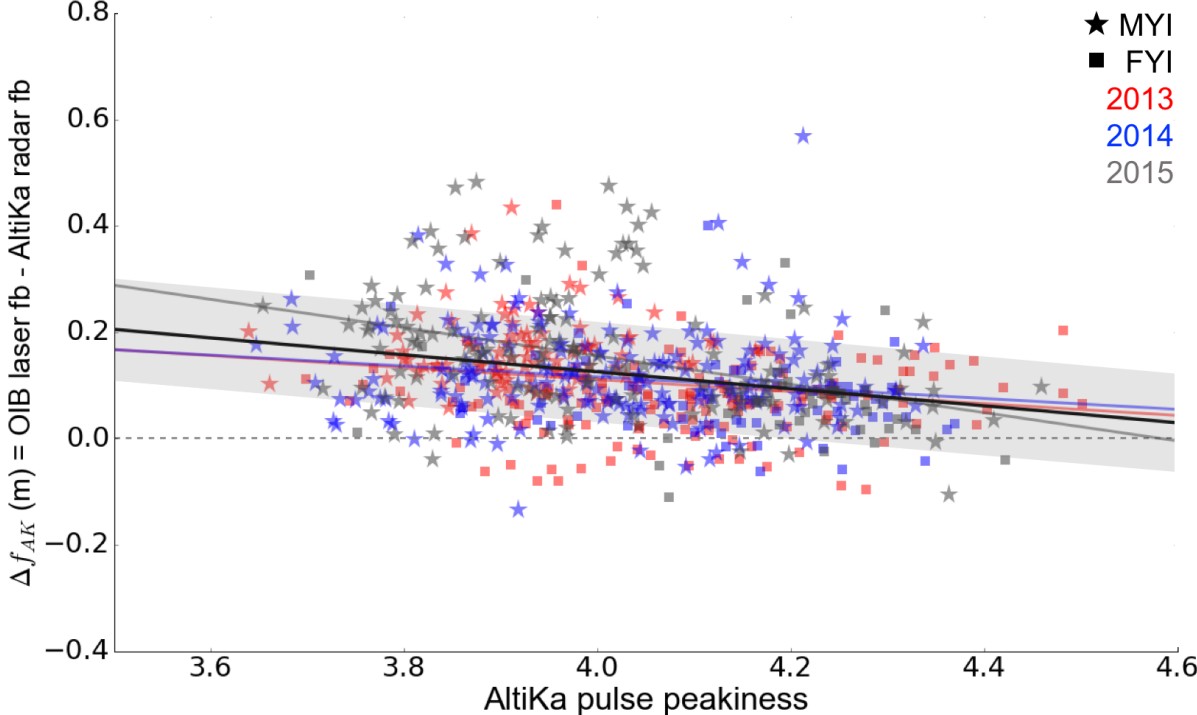

**Figure 1.** $\Delta f_{AK}$, defined as OIB laser freeboard minus AltiKa radar freeboard, plotted against AltiKa Pulse Peakiness, for the OIB spring campaigns of 2013 (red), 2014 (blue) and 2015 (grey). Multi-year and first-year ice are plotted with stars and squares respectively. The combined (all years) linear regression fit (CLRF), shown by the black line, has slope of -0.16 and intercept of 0.76. The shaded area around the CLRF shows the 68% prediction interval, corresponding to a standard error (SE) on $\Delta f_{AK}$ of 9.4 cm.

over rough ice, in support of the assumption that i) rough, multi-year ice has a thicker snow cover and ii) seasonal ice is likely subject to brine wicking which prevents radar propagation through the snow (Nandan et al., 2017). Ultimately we cannot separate the influence of individual sources of bias and therefore these observations are purely speculative.

### 3.6 CS-2 calibration with Operation IceBridge

5   The procedure for calibrating CS-2 with OIB is identical to that outlined above for AltiKa, but here $\Delta f_{CS}$ is defined as OIB radar freeboard (see Sect. 3.4) minus CS-2 radar freeboard. $\Delta f_{CS}$ plotted against CS-2 PP is shown in Fig. 2. The CLRF, shown by the black line, has a slope of 0.07 and negative intercept of -0.51. As before, the shaded area around the CLRF shows the 68% prediction interval, and corresponds to a $\pm 7.5$ cm uncertainty (1 Standard Error) on $\Delta f_{CS}$. Since CS-2 has better coverage over the pole, there are more data points retrieved for CS-2 (1423 as opposed to 656 for AltiKa), giving its regression
10  smaller prediction intervals.

   For most of CS-2's PP range (up to ∼7), the CLRF is negative. It is most negative at lower PP, indicating that CS-2's freeboard lies higher above the snow/ice interface over rough ice. This is in agreement with rougher ice exhibiting a thicker





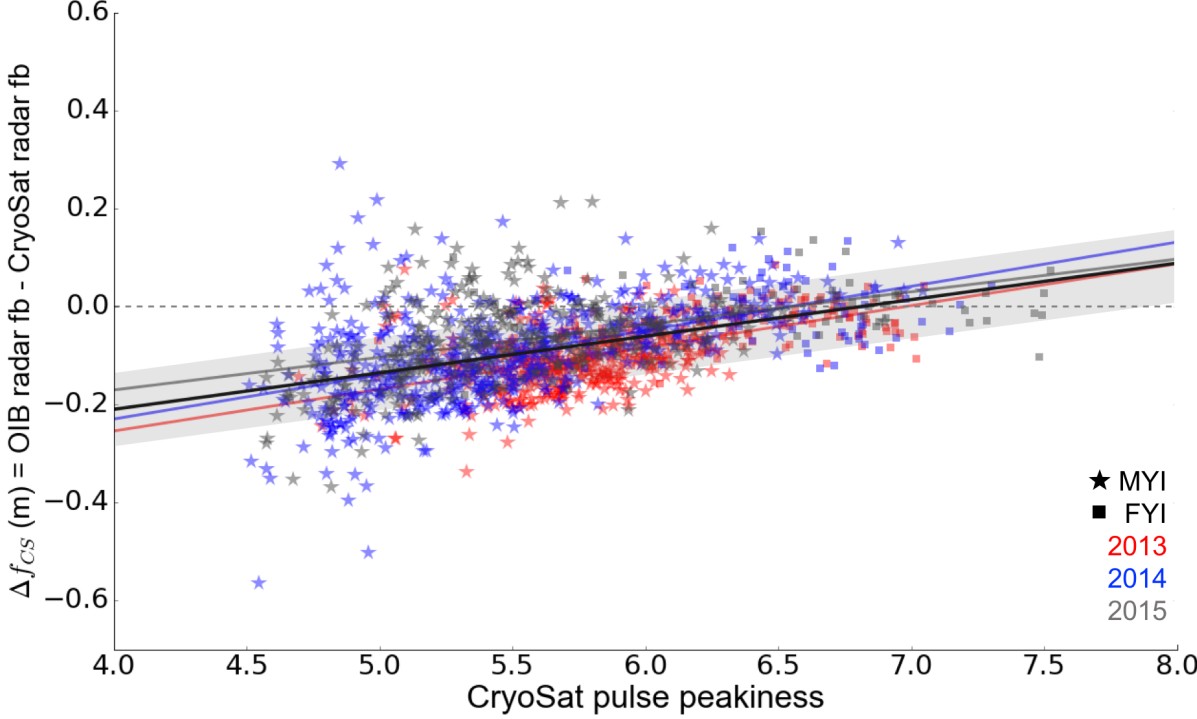

**Figure 2.** $\Delta f_{CS}$, defined as OIB theoretical radar freeboard minus CS-2 radar freeboard, plotted against CS-2 Pulse Peakiness, for the OIB spring campaigns of 2013 (red), 2014 (blue) and 2015 (grey). Multi-year and first-year ice are plotted with stars and squares respectively. The combined (all years) linear regression fit (CLRF), shown by the black line, has slope of 0.07 and intercept of -0.51. The shaded area around the CLRF shows the 68% prediction interval, corresponding to a standard error (SE) on $\Delta f_{CS}$ of 7.5 cm.

snow cover and the radar pulse therefore being limited from getting as near to the snow/ice interface as where the snow is thinner. Above PP of 7, the CLRF becomes positive, suggesting that CS-2's freeboard lies below the snow/ice interface for smooth ice. We do not expect CS-2's pulse to penetrate into the ice pack, and attribute this to a poor fit of the linear regression to data points with peakiness above ~7.

5 ## 4 Results

### 4.1 Case Study November 2015 to April 2016

To derive snow depth, along-track freeboard measurements for AltiKa and CS-2 are calibrated as a function of PP according to the combined linear regression fits (CLRFs) derived in the previous section, and then averaged onto a 1.5° longitude by 0.5° latitude monthly grids. A finer grid resolution than for the calibration analysis is afforded given the coverage of one

10 month's worth of data as compared to the 21 days (±10 days window) averaged previously. The calibrated CS-2 freeboard is subtracted from calibrated AltiKa freeboard, and multiplied by a factor $c_s/c$ = 0.781 to convert to snow depth. Figure 3



summarises the retrieved monthly Dual-altimeter snow thicknesses (DuST) from November 2015 to April 2016, smoothed using a Gaussian convolution filter with a standard deviation of 30 km. The delineation of multi-year and first-year ice is shown by the dashed black lines, adapted from OSI SAF Quicklook daily sea ice type maps for the 15th day of each month, available at http://www.osi-saf.org.

**Figure 3.** Monthly snow depths for the growth season November 2015 (top left) to April 2016 (bottom right), derived from AltiKa minus CS-2 calibrated freeboard, smoothed using a Gaussian convolution filter with a standard deviation of 30 km. The multi-year ice boundary for each month is shown by the dashed black line, adapted from the OSI SAF Quick look sea ice type map for the 15th day of the month, available at http://www.osi-saf.org/.





Spatial distribution of snow depth follows the expected pattern of a thin snow cover over seasonal ice (up to 20 cm) and thicker snow over multi-year ice (30-40 cm) (Warren et al., 1999), which in recent years is limited to regions north of the Canadian Archipelago (CAA) and Greenland, and the Fram Strait. However, seasonal deposition of snow occurs between November and April, corresponding with the locations of predominant cyclone tracks in winter (e.g. the Aluetian Low on the

Pacific side, and the North Atlantic Storm tracks). In particular, snow predominantly accumulates within the Chukchi Sea, and within the Kara, Barents and East Greenland Seas. As well as precipitation events, ice drift governs snow distribution through the advection of snow-loaded sea ice parcels around the ocean. Therefore in order to understand the seasonal evolution of the snow cover, we compare snow depth maps with monthly sea ice motion vectors from the National Snow and Ice Data Centre (NSIDC, available at https://daacdata.apps.nsidc.org), shown in Fig. 4. We expect snow accumulation west of Banks Island

in the CAA is the result of westward transport of multi-year ice by the Beaufort Gyre. Snow depths in the Kara Sea appear high given the advection of ice out of this region throughout the season, however we cannot rule out anomalous precipitation events. Typically 20-40 extreme cyclones occur each winter within the North Atlantic, but in recent years there has been a trend towards increased frequency of cyclones, particularly near Svalbard (Rinke et al., 2017). These cyclones, while they transport heat and moisture into the Arctic and may impact the sea ice edge location (Boisvert et al., 2016; Ricker et al., 2017), can also

be associated with increased precipitation. At the same time it is important to note that OIB only operates in the Western Arctic and therefore the Siberian seas are unconstrained by observations which may lead to erroneous snow depths.

To understand where greatest accumulation of snow occurs over the season, we also plot the difference between November 2015 and April 2016 snow depth in Fig. 5. Snow accumulation is highest in the Western Beaufort sea, in particular adjacent to the coast of Canada. We attribute this to the advection of snow-loaded multi-year ice by the Beaufort Gyre, supported by the

visible shift of the multi-year ice boundary through the season (Fig. 3). Accumulation also occurs in the Fram Strait, which we expect to be the result of southward advection of multi-year ice from the central Arctic Ocean in December and April, as well as snow deposition from the North Atlantic Storm tracks. High accumulation in the southern Chuckchi Sea could also be explained by strong advective currents pushing snow-loaded ice into this area, particularly from November to January, as well as snow precipitation from the Aleutian Low. Negative snow depth changes are generally small, and are predominantly visible

in the centre of the Beaufort and Laptev Seas. In accordance with Fig. 4 we expect these negative accumulations to be the result of advection transporting snow-loaded ice parcels out of these regions and perhaps new ice formation.

Since OIB campaigns only operate in the western Arctic Ocean, north of the CAA and in the Lincoln and Beaufort Seas, no observations from the eastern Arctic go into our calibrations. Thus, the calibration functions derived are unconstrained outside of this area and we have less confidence in the snow depths in the eastern Arctic. Further, the calibration relationships are only

strictly valid in spring, when OIB operates, so caution is warranted in using these products for seasonal variability of snow depth analysis.

A secondary limitation is the large data gap associated with AltiKa's upper latitudinal limit of 81.5°. This region contains a large proportion of the Arctic's thick multi-year ice and thus observations of snow depth could provide valuable insight as the icepack transitions from multi-year to first-year ice. Furthermore, for a snow depth product to be useful for integration into

sea ice thickness retrievals as discussed in the introduction, one that extends to CS-2's latitude range up to 88° is desirable.



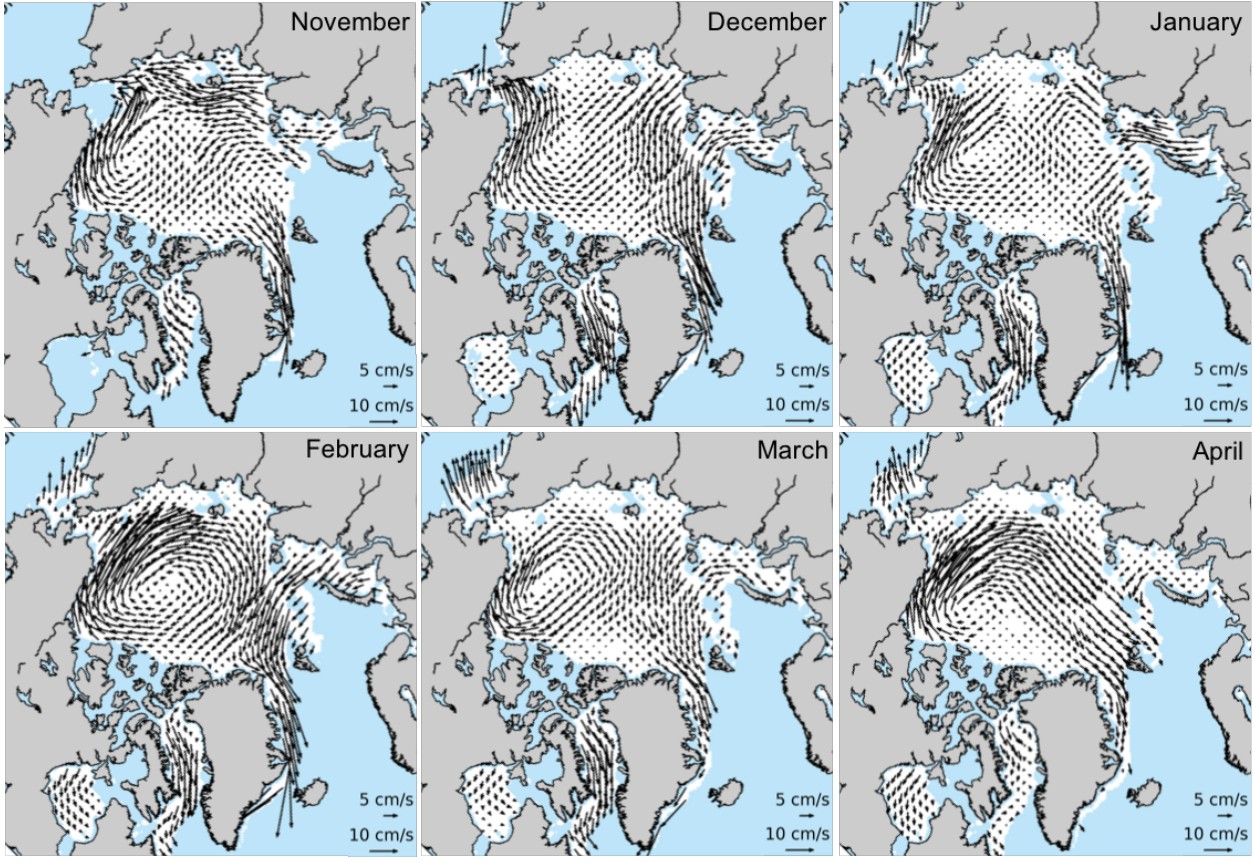

**Figure 4.** NSIDC November 2015 to April 2016 monthly mean sea ice drift vectors. Adapted from images retrieved from https://daacdata. apps.nsidc.org/pub/DATASETS/nsidc0116_icemotion_vectors_v3/browse/north/.

Alternatively, dual-frequency operation from the same satellite platform would open the potential for snow depth retrievals along the satellite track.

## 4.2 Error calculation

The equation for calculating snow depth, $h_s$, by our methodology is:

$$h_s = 0.781^* \Big( (f_{AK} + \Delta f_{AK}) - (f_{CS} + \Delta f_{CS}) \Big) \tag{5}$$

Where $f_{AK}$ and $f_{CS}$ are AltiKa and CS-2 freeboard and $\Delta f_{AK}$ and $\Delta f_{CS}$ are the AltiKa and CS-2 freeboard corrections (see Sects. 3.5 and 3.6)

From propagation of errors on Eq. (5), the uncertainty on snow depth, $\sigma_{h_s}$, is given by:

$$\sigma_{h_s} = 0.781 * \sqrt{\sigma^2_{f_{AK}} + \sigma^2_{\Delta f_{AK}} + \sigma^2_{f_{CS}} + \sigma^2_{\Delta f_{CS}}} \tag{6}$$





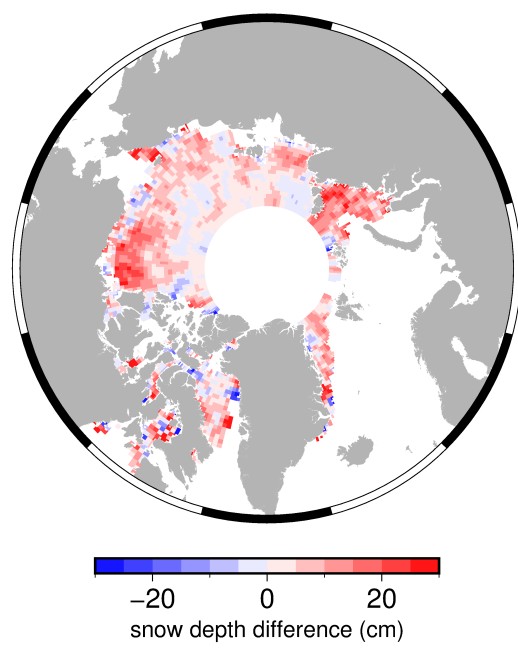

**Figure 5.** April 2016 minus November 2015 DuST snow depth.

The 68% prediction intervals, represented by the shaded areas in Figs. 1 and 2, provide a $\pm 1$ Standard Error (SE) estimate on $\Delta f_{CS}$ of 7.5 cm and $\Delta f_{AK}$ of 9.4 cm.

Since our snow product is monthly-gridded we are interested in monthly-gridded snow depth uncertainty. Therefore $\sigma_{f_{AK}}$ and $\sigma_{f_{CS}}$ are the errors on the monthly-gridded satellite freeboards to which the calibration corrections are being applied.

5  Tilling et al. (2017) provide an estimate of monthly-averaged freeboard error for CS-2, for their grid, of 2 cm. This is dominated by uncertainty on sea surface height estimation, which they calculate to have a standard deviation of 4 cm. Sea surface elevation is calculated from waveforms identified as leads (see Sect. 3.3). Lead elevations within a 200 km along-track window about each floe measurement are fit with a linear regression to estimate the sea surface elevation beneath the floe and thus calculate the freeboard. As such, along-track floe measurements are not decorrelated at length scales less than 200 km and sea surface

10  uncertainty is not reduced from grid-cell averaging of data from the same satellite pass. Since the interpolation is performed along-track, separate satellite passes over each grid cell over the month *are* decorrelated, and thus the error is minimised by $1/\sqrt{N}$, where $N$ is the number of passes over a grid cell in one month. Tilling et al. (2017) calculate that for their grid N averages 4 or more.

For our chosen snow depth grid of $1.5°$ longitude by $0.5°$ latitude, we evaluate monthly along-track CS-2 data in order to

15  quantify the number of passes per grid cell. Due to the diverging of satellite tracks with decreasing latitude, we find N varies between $\sim 2$ at lower latitudes ($70°$) to $\sim 5$ at our highest latitude of $81.5°$. This results in a reduction of the 4 cm standard deviation on sea surface height to an error between $\sim 3$ and 1.8 cm, latitude depending.





Since the same 200 km along-track window is applied during AltiKa freeboard processing, we similarly assign a 4 cm standard deviation on sea surface height retrieval for AltiKa. However, AltiKa has many more passes than CS-2 between 70 and 81.5 since it does not survey the pole. For AltiKa we find average monthly passes per grid cell vary from ∼5 at 70° to ∼40 at 81.5°. This results in a reduction in sea surface interpolation error to 1.8 cm and 0.6 cm respectively. As a conservative

estimate of the error on our snow depth product, we assign $\sigma_{f_{AK}}$ and $\sigma_{f_{CS}}$ values of 1.8 and 2.8 cm respectively, giving a final error on gridded snow depth $\sigma_{h_s}$ of 9.7 cm, from Eq. (6).

The main contribution to snow depth error is the prediction intervals from the calibration functions (see Sects. 3.5 and 3.6). This uncertainty could be reduced with the addition of more data points, i.e. more seasons of coincident satellite and OIB measurements. At time of publication OIB data for spring 2017 had not been made publicly available.

Since during comparison of satellite and OIB laser and radar freeboard we discarded grid cells with less than 50 points, we expect random error on OIB freeboards to be minimised such that they will not dominate the final snow depth uncertainty. However, any systematic bias that exists on these products due to processing technique or measurement error will impact the calibration functions and therefore our final snow depth retrievals. An example of a systematic bias would be the false detection of air/snow and snow/ice interfaces from snow radar data. A recent study by Kwok et al. (2017) found that different research

groups' treatment of returns in order to locate these interfaces are not consistent, leading to different snow depth estimates from the same data. In this study we have only used the OIB Quick Look data set since this was what was available at the time, in particular given our need for coincident ATM laser freeboard in order to find radar freeboard. We acknowledge however that it would be worthwhile to investigate the difference on DuST snow depth retrievals using snow radar data processed by other research groups in order to asses the sensitivity of our product.

## 5   Discussion

### 5.1   Comparison with Operation IceBridge

We compare snow depth retrieved by our methodology with OIB snow depths from Spring 2016 following the same procedure outlined in Sects. 3.5 and 3.6. For each day of the 2016 campaign, OIB snow depths are averaged onto the 2° longitude x 0.5° latitude grid and grid cells containing less than 50 individual points are discarded to remove speckle noise, as before. Calibrated

AltiKa and CS-2 freeboards for the ±10 days surrounding the campaign day are averaged onto the same grid and grid cells with less than 50 AltiKa or CS-2 points are discarded. Gridded calibrated CS-2 freeboard is subtracted from gridded calibrated AltiKa freeboard and multiplied by factor $c_s/c = 0.781$, as previously. The resulting snow depth grid is then interpolated at the average position of the OIB data within each valid OIB grid cell. The Dual-altimeter Snow Thickness (DuST) retrieved for each point is plotted against OIB snow depth, shown in Fig. 6(b). We find a root-mean-square deviation (RMSD) of 7.6 cm

and a mean difference of 2.1 cm. The linear regression fit, shown by the dashed black line, yields a correlation coefficient of r = 0.73. Figure 6(a) shows OIB snow depth plotted over DuST for the corresponding period.

Since OIB data from 2013-2015 was used to calibrate the satellite freeboards, this cannot be considered a true validation exercise. However, if OIB is considered as providing accurate snow depth estimates, then the 2016 comparison with our product





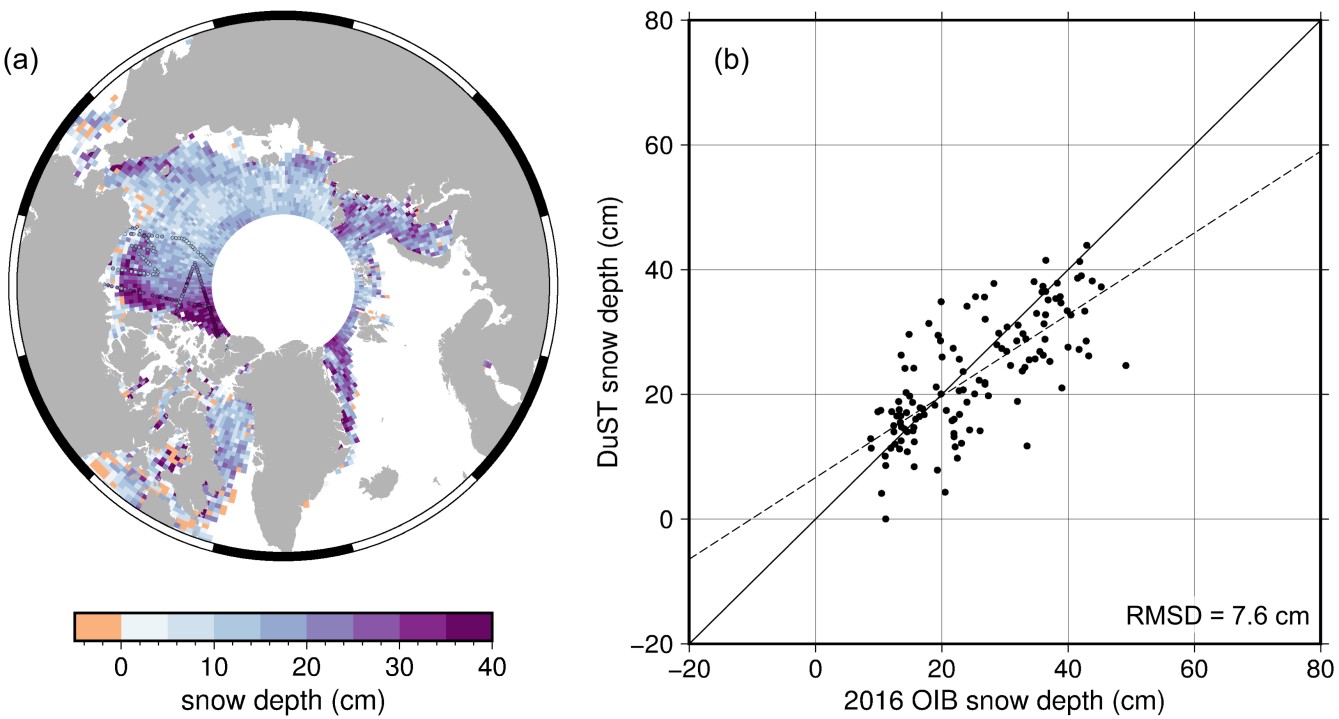

**Figure 6.** Comparison between OIB snow depth data for the 2016 campaign season and DuST snow depth. Satellite data for the ±10 days around each OIB campaign day is corrected according to the derived calibration functions. Gridded calibrated AltiKa minus calibrated CS-2 is then sampled at the average position of OIB data within each grid cells to retrieve a corresponding Dual-altimeter snow thickness (DuST). (a) OIB snow depth plotted over DuST for the ±10 days around each OIB campaign day. (b) OIB snow depth vs. DuST. We find a RMSD of 7.6 cm and mean difference of 2.1 cm. The linear regression fit, shown by the dashed black line, yields a correlation coefficient of r = 0.73

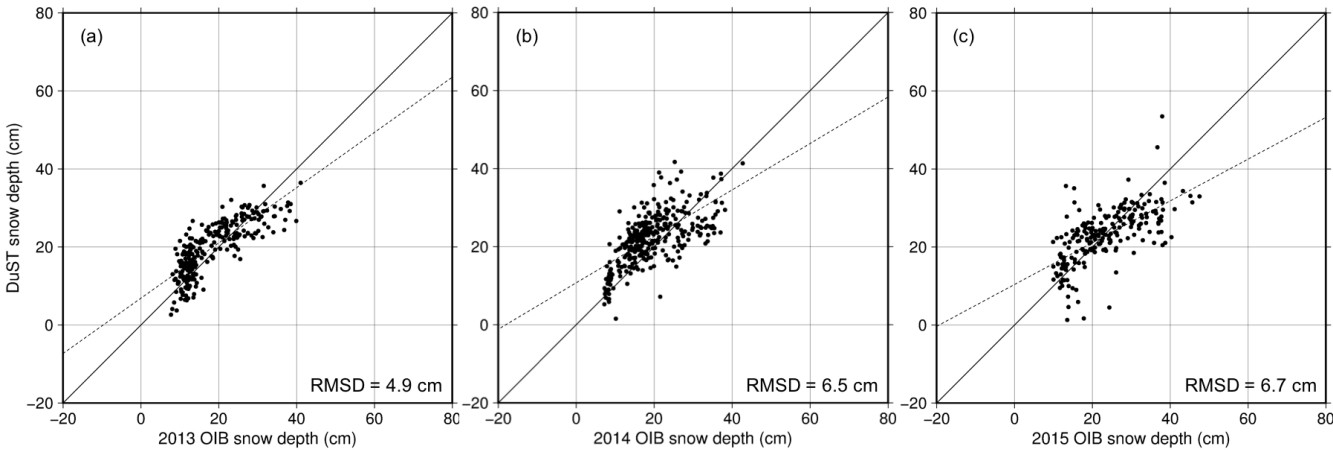

**Figure 7.** Comparison of DuST and OIB snow depths for the a) 2013 b) 2014 and c) 2015 spring campaigns. Statistical results for all years are summarised in Table 2.



**Table 2.** Results of OIB and DuST comparison for the years 2013-2016.

|  | 2013 | 2014 | 2015 | 2016 |
|---|---|---|---|---|
| Root-mean-square deviation (RMSD) | 4.9 cm | 6.5 cm | 6.7 cm | 7.6 cm |
| Difference in means | 1.2 cm | 2.6 cm | 0.3 cm | 2.1 cm |
| Correlation coefficient, r | 0.80 | 0.65 | 0.64 | 0.73 |

implies both the ability to upscale OIB snow depths to the wider Arctic and use the derived calibration relationships for future snow depth estimates (e.g. when OIB is no longer operational).

The original analysis outlined in Sects. 3.6 and 3.5 was repeated, successively omitting each of the 2013-2015 OIB seasons to derive calibration functions and generate snow depths for the omitted year. DuST snow depths were then compared against OIB snow depths by the method outlined above. Results are shown in Fig. 7 and summarised in Table 2.

## 5.2 Application of DuST to ICESat-Envisat

The methodology outlined above demonstrates the ability to calibrate satellite freeboards with an independent data set in order to derive snow depth. It can be applied to any freeboard data sets and could be usefully applied once ICESat-2 is launched later this year. In view of this possibility, we have applied the methodology to a comparison of ICESat and Envisat, whose periods of operation overlapped between 2003 and 2009.

The Radar Altimeter 2 (RA2) instrument operated on the Envisat satellite from 2002 until 2012. It was a pulse-limited Ku-band radar altimeter which like SIRAL, operated at a central frequency of 13.575 GHz. NASA's ICESat mission featured a Geoscience Laser Altimeter System (GLAS) in order to accurately measure changes in the elevation of the Antarctic and Greenland ice sheets. This laser was also used to estimate ice thickness from laser freeboard retrieval (e.g. (Kwok et al., 2007)). Between 2003 and 2009, ICESat completed 17 observational campaigns; once every spring (Feb/March) and autumn (Oct/November) as well as three in the summers of 2004, 2005 and 2006.

ICESat had a 70 m diameter footprint, so we assume that biases due to footprint size or retracking method are negligible and that it offers accurate estimates of the snow freeboard. We use available ICESat freeboard data (version 1) from NSIDC, (Yi and Zwally, 2009), in our analysis. Envisat freeboard data were processed by CPOM and the reader is referred to Ridout and Ivanova (2013) for further details on the algorithm.

Following the procedure outlined in Sect. 3.5, Envisat freeboard is calibrated to the snow/ice interface. Envisat has a larger footprint than AltiKa due to its lower operating frequency and higher bandwidth, nominally 2-10 km diameter (Connor et al., 2009). As such, the waveform returns are more often classified as ambiguous (showing a complex mixture of scattering behaviour) and discarded, as discussed with reference to AltiKa in Sect. 3.3. As a result, Envisat data are sparsely populated and in order to have sufficient coverage for comparison with OIB data and 50 or more points per grid cell (to reduce speckle noise), it was necessary to increase both the grid resolution and time window as compared with the calibration procedure performed for AltiKa and CS-2.




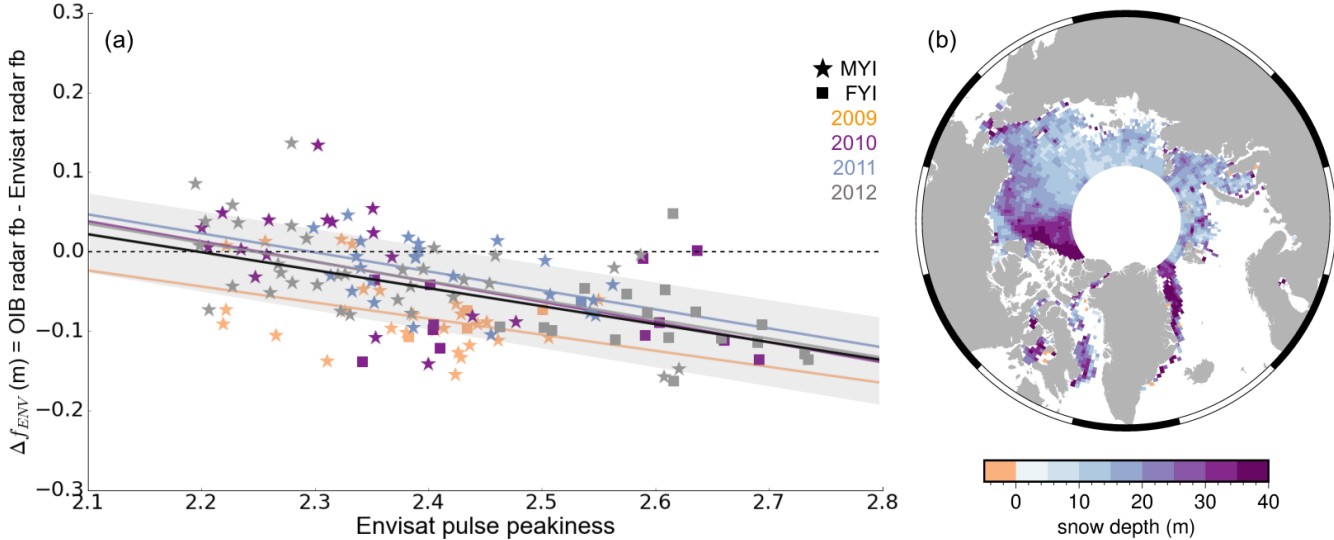

**Figure 8.** (a) Envisat calibration relationship, derived from comparison of coincident OIB and Envisat data. Data and corresponding linear regression fits for 2009, 2010, 2011 and 2012 and shown in orange, purple, blue and grey respectively. Star and square symbols represent multi-year and seasonal ice respectively. (b) Snow depth for ICESat's '3E' laser period (22nd February 2006 to 27th March 2006), retrieved by subtracting calibrated Envisat freeboard from ICESat freeboard and multiplying by a factor 0.781.

Satellite data for the $\pm 15$ days surrounding each 2009-2012 OIB campaign day were averaged onto a onto a 3°longitude x 0.75°latitude grid. $\Delta f_{ENV}$, defined as OIB radar freeboard minus Envisat freeboard, plotted against Envisat PP is shown in Fig. 8(a). Data from 2009, 2010, 2011 and 2012 and their corresponding linear regression fits are plotted in orange, purple, blue and grey respectively to demonstrate year to year consistency. As before, multi-year and first-year ice are distinguished

5 by star and square markers in order to illustrate the variation of PP with ice type. The combined (all years) linear regression fit (CLRF) is shown by the black line and has slope of -0.23 and intercept 0.50. The shaded area shows the 68% prediction interval about the CLRF, corresponding to a $\pm 5$ cm standard error (SE) on $\Delta f_{ENV}$.

Dual-altimeter Snow Thickness (DuST), retrieved by subtracting calibrated Envisat freeboard from ICESat freeboard is shown in Fig. 8(b) for the ICESat laser period '3E' (22nd February 2006 to 27th March 2006). Snow depth agrees with expected

10 spatial distribution and magnitude, with thicker snow (30+ cm) over multi-year ice to the north of the Canadian Archipelago and in the Fram Strait, and thinner snow cover (< 20 cm) over seasonal ice. Overall higher magnitudes as compared with March 2016 (Fig. 3) could be the result of a decline in multi-year ice fraction and precipitation over the past decade. Though validation is required, the result demonstrates the viability of combining laser and calibrated radar freeboard to retrieve snow depth.



## 6    Conclusions

Using independent snow and ice freeboard data from OIB, we derived calibration relationships to align AltiKa to the snow surface and CS-2 to the ice/snow interface, as a function of their pulse peakiness. Calibrated CS-2 and AltiKa freeboard data were then combined to generate spatially extensive snow depth estimates across the Arctic Ocean between 2013 and 2016.

The Dual-altimeter Snow Thickness (DuST) product was evaluated against OIB snow depth by successively omitting each year of OIB data from the calibration procedure, returning root-mean-square deviations of 4.9, 6.5, 6.7 and 7.6 cm for the years 2013, 2014, 2015 and 2016 respectively. While the OIB snow depth data cannot be considered statistically independent validation of the DuST product, this evaluation does demonstrate the ability to up-scale OIB snow depths to the wider-Arctic. However, the DuST snow depth estimates remain unconstrained and unevaluated outside of the Western Arctic and the spring
season, due to a lack of coincident data.

A more thorough validation using ice mass balance buoys as well as comparisons with other derived snow products is the subject of future work. Looking further ahead, the upcoming MOSAiC ice drift campaign in autumn 2019 will provide a unique opportunity for validation in regions not sampled by OIB (e.g. the eastern Arctic) and throughout a full annual cycle. A dedicated dual-radar study is planned during the MOSAiC experiment, using in-situ and on-aircraft Ku-Ka band radar to
quantify radar backscatter at each frequency together with snow depth and ice thickness measurements. This in conjunction with AltiKa and CS-2 observations will provide valuable insight into the validity of our calibration functions and retrieved DuST snow depths.

Our methodology can also be applied to retrieve snow depth from coincident satellite radar and laser altimetry, which will have particular relevance when ICESat-2 is launched (scheduled late 2018). Here, we demonstrated the applicability of the
method to the ICESat and Envisat satellites, offering promising potential for the future retrieval of snow depth on Arctic sea ice from CS-2 and ICESat-2, with better coverage over the pole.

*Competing interests.*    The authors declare that they have no conflict of interest.

*Acknowledgements.*    This work was funded primarily by a National Environmental Research Council Doctoral Training Partnership grant (code 'NE/L002485/1') and in part by the Arctic+ European Space Agency snow project ESA/AO/1-8377/15/I-NB NB - 'STSE - Arctic+'.
The authors wish to thank Richard Chandler, UCL, for help in preparing this manuscript. T. Armitage was supported at the Jet Propulsion Laboratory, California Institute of Technology, under a contract with the National Aeronautics and Space Administration. CryoSat-2 and Envisat data were provided by the European Space Agency, and processed by the Centre for Polar Observation and Modelling. AltiKa data were provided by AVISO. ICESat freeboard, Operation IceBridge data and sea ice motion vectors were provided by the National Snow and Ice Data Centre. Sea ice type masks were provided by the Ocean and Sea Ice Satellite Application Facilities.



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
