# Peer review of "Estimating snow depth over Arctic sea ice from calibrated dual-frequency radar freeboards"

_The Cryosphere, 2018_

## Referee Comment (RC1) · 12 Apr 2018

Review for "Estimating snow depth over arctic sea ice from calibrated dual-frequency radar freeboards" by Lawrence et al.

General comments:

The study "Estimating snow depth over arctic sea ice from calibrated dual-frequency radar freeboards " by Lawrence et al. uses a combination of altimeters operating at different frequencies (Ku and Ka bands) and flying over the Arctic at the same period (2013-2016) in order to estimate snow depth at the top of sea ice.

Based on previous studies, the authors consider that the main return of the Ka-band radar signal arises from an upper part of the snowpack and that the main return of

the Ku-band radar signal originates from a lower part of the snow pack. Using this difference of penetration depth into the snowpack, they estimate snow depth at the top of sea ice by calculating the difference of freeboard height between SARAL/AltiKa (Ka-band) and CryoSat-2 (Ku-band). Before processing the freeboard difference, the authors correct freeboard biases related to radar penetration/surface state. To correct these biases, they fit their Ka and Ku freeboard measurements using laser and radar measurements from the Operation Ice Bridge (OIB) 2013-2016 campaigns. To validate their "Dual-altimeter Snow Depth" estimates, the authors use "independent" snow depth measurements from the Operation Ice Bridge airborne campaigns. Further, they show that the methodology derived with CryoSat-2 and AltiKa can be reproduced using Envisat (Ku-band) and ICESat (Laser).

The paper focuses on a very relevant topic as snow depth is one of the most important sources of uncertainties when converting ice freeboard to ice thickness. Hence, measuring snow depth at pan-arctic scale with a good temporal resolution could strongly help to improve current sea ice thickness estimates. In addition, snow depth is a key thermodynamics parameter as it isolates sea ice from the cold atmosphere in winter and reflects an important amount of solar radiations in summer. Being able to measure snow depth at large scale could therefore truly help to improve our understanding of sea ice growth and melt processes.

In my opinion, the approach of using a combination of altimeter measurements to estimate snow depth deserves publication. However, I have some major remarks that must be addressed before the paper can be published:

1) While the authors considers that the Ka and Ku radar signals do not penetrate identically into the snowpack, it is not clearly stated where the main returns arise from. The authors quote Armitage and Ridout (2015) and Guerreiro et al. (2016), which draw different conclusions, but they don't clearly give their thoughts. This precision is crucial as one needs to know if the freeboard fit they perform with OIB is used to correct footprint effects or/and penetration effects.

The authors say "Freeboard estimates from CryoSat-2 (Ku-band) and AltiKa (Ka-band) are calibrated against data from NASA's Operation IceBridge (OIB) to align AltiKa to the snow surface and CryoSat-2". Considering this sentence, I assume that they consider (as in Armitage and Ridout (2015)) that Ku does not fully penetrate the snowpack while Ka does penetrate it a little bit, right? If yes, this raises an important question that should be answered more clearly: why the penetration of the Ka and Ku-bands would change from one area to another (Figure 1 and 2 show that the corrections are not constant)? Also, this assumption seems to not take into account the results shown in Kurtz et al. (2014) and Guerreiro et al. (2017), why?

2) The "error calculation section" (4.2) deserves some improvements. First of all, the authors calculate the uncertainty from an error propagation using a quadratic formula with variables that are clearly not independent. The variables covariance should be taken into account to avoid this issue. Also, they consider that AltiKa and CryoSat-2 have a similar standard deviation on sea surface estimate and they come to the conclusion that, since AltiKa coverage is better than CryoSat-2, AltiKa freeboard error is smaller than that of CryoSat-2. In my opinion, this cannot be true even with the better coverage of AltiKa in the studied region. To derive appropriate errors, the authors should calculate the standard deviation on sea surface for each satellite mission before injecting it in equation (6). Finally, I am not sure what the authors mean by "correlation coefficient" in section 5.1. According to the values in Table 2, I am guessing that they calculate the fit regression line slope. I think it would be preferable to provide a Pearson coefficient R, which is a more common parameter.

3) I acknowledge that contemporary large-scale snow depth measurements are extremely rare and that using the same dataset for calibration and validation is one of the only existing options. Having said that, I would suggest to modify the plan of section 5.1 by not considering the year 2016 as a particular one (Figure 6). At the end of the day, Figure 6 (2016) and Figure 7 (2013-2014-2015) are almost identical: you remove observation from the considered year to evaluate your DuST snow depth. Thus, it does

not require two figures nor two comments/conclusions.

Minor comments:

Page 1 L 4-6: "Freeboard estimates . . . ice/snow interface". Does it mean that Ka/Ku don't stop at the air-snow/snow ice interface?

L 23: As you mention Envisat above, you should also quote Giles et al. (2008).

Page 2: L15-18: This is arguable. For LRM altimeters, the uncertainty related to freeboard height is at least as large as the one related to snow depth.

Page 3: L 31: To be more precise, Guerreiro et al. (2017) suggest that the Ka-band signal stops within the first few centimetres and that the Ku-band signal can stop before the snow-ice interface in case of large snow grains.

L33: This is not exact: The first study that showed AltiKa freeboard measurements was the one by Maheshwari et al. (2015).

Page 4:

L13: Here and elsewhere, can you mention which footprint you talk about (beam-limited or pulse-limited).

L15: So here, you choose to follow the conclusion given in Armitage and Ridout (2015), which is that the Ka and Ku signals stop within the snow pack, right? If yes, could you state it more clearly? Also, considering the literature you quoted (or not) (Kurtz et al., 2014; Maheshwari et al., 2015; Guerreiro et al., 2016; Schwegman et al., 2015), could you please explain this choice. This is indeed a major point as your entire study is based on this assumption.

L29: Not exact: see my previous comments

Page 6:

L3: In Armitage and Ridout (2015), I believe that the authors follow another condition

related to the Leading Edge Width (see supplementary material). Could you check on that please?

L26-31: To me, this way to proceed raises an important question: As you mention it above, the altimeter range can be biased by waveform hooking due to the proximity of specular reflections. Thus, if you calibrate your freeboards in a particular region (the one overflown by OIB for example), the calibration will likely depend on the density of Off-Nadir reflections found in this region. Consequently, the derived calibration might not work in regions where the density of Off-Nadir reflections is different. To check if your calibration is region-dependent or not, a simple test can be operated: you can plot the residuals of Figures 1 and 2 on a map and check if you observe regional patterns or not. This figure could be provided in the supplementary material.

Page 7:

L16-19: How do you evaluate the spatial and temporal resolution?

L28-30: Could you give the correlation coefficient (Pearson's)?

Page 9:

L2-4: As you consider that the bias you fit is due to penetration effects, then yes, a $\Delta fs > 0$ would imply that the Ku-band signal penetrates through sea ice. However, if one considers that this bias is also due to surface properties (roughness for example), positive $\Delta fs$ values would simply suggest that the empirical retracking you use is not adapted to sea ice surfaces. This was clearly demonstrated in the study by Kurtz et al. (2014). Could you provide with a more detailed comment by integrating this other aspect?

Page 11:

L-15-16: As you do not clearly mention why you need to calibrate AltiKa and CryoSat-2 freeboards (penetration depth? surface properties?, ...), this conclusion is hard to understand. Why would your calibration be different from on region to another?

Because of snow properties? Lead density? Surface diffusivity? You need to give more details in order to provide a more convincing conclusion.

L29-31: Same remark as above.

Page 12:

L3: Shouldn't the title be "Uncertainty calculation"? An error should be relative to a "truth measurement"...

Eq 6: As fAk and $\Delta$fAk are clearly not independent (see Figure 1), you must take into account their covariance to calculate the uncertainty.

Page 14:

L1-2: Why do you apply the same standard variation value as for CryoSat-2 (4 cm)? As far as I know AltiKa sea level standard deviation is much larger than that of CryoSat-2. I would recommend to re-calculate a standard deviation for the two datasets here in order to make sure you have the right values.

Page 15: Figure 6: I am quite surprised about the r value you provide (0.73) considering the figure you show. How do you calculate this coefficient? It seems to me that you provide with the fit regression line slope. Am I right? If yes, I think it would be preferable to provide a Pearson coefficient R, which is more common parameter.

Page 16:

Table 2: Same remark as above.

Section 5.1: I don't understand why you consider 2016 as a particular year. As suggested by Table 2, the comparison for 2016 is almost identical as for the other years (except that you don't use 2016 to calibrate your DuST snow depth for the 2013-2015 periods). In my opinion you should not make any distinction between 2013-2015 and 2016 and re-write this section as such.

L22: There is no link between the footprint size and the bandwidth. Also, you can have a similar footprint with 2 different frequencies depending on the antenna size.

Page 17:

L1: "Onto a" is written twice

L3-5: This description should be moved into the figure caption.

L10: What does 30+ mean? Can you provide with a range of values instead?

Page 18:

L19: Considering that you did not use validation data to validate your results, I would not use "demonstrated" here...
* * *

---

## Referee Comment (RC2) · Anonymous Referee #2 · 29 May 2018

The paper "Estimating snow depth over Arctic sea ice from calibrated dual-frequency radar freeboards" by Lawrence et al. deals with estimating snow depth by combining satellite-based measurements of snow and ice freeboard. The method requires prior calibration with independent freeboard measurements. Here, CryoSat-2 and AltiKa satellite freeboard measurements are calibrated with airborne Operation IceBridge (OIB) measurements.

The latter raises one of my main concerns: The method, as presented here, relies on having reliable independent freeboard data, which at the moment is only provided by OIB data. However, there is disagreement within the science community on how to interpret the OIB radar measurements, i.e. different retrieval algorithms differ in the way air-snow and snow-ice interfaces are detected and localized. A recent paper by Kwok et al. (2017) showed that this caused OIB snow depths as retrieved from different groups to differ on average by up to 7 cm (for first-year ice) and 12 cm (for multi-year ice) for the 2013-2015 data (see Fig. 7 & 8 in Kwok et al., 2017), which is used in the paper presented here. The variability of snow depths is also quite different (so it is not just a constant bias between the different products). Though this problem is briefly mentioned in the paper presented here, this is only done rather late (on p. 14, l. 14), which does not represent how severe this issue is for the proposed method of retrieving snow depth. I think that this should be mentioned and discussed far earlier and with more emphasis because it has major implications on the usability and accuracy of the proposed retrieval method! Ideally, the authors would perform their comparison not only for OIB quicklook data, but also for (at least) one of the other OIB-based freeboard retrieval data sets to estimate how much this can influence the results.

A further concern is that the study of Guerreiro et al. (2016) also uses CryoSat-2 and AltiKa freeboard measurements to retrieve snow depth. Instead of calibrating these Ku and Ka-band measurements with independent data (as done here), they theoretically analyze the penetration depths of both radar altimeters in snow and use snow density estimates to modify the Ku-band radar signal's velocity through the snow. In their study, they compare their retrieved snow depths with OIB snow depths for the same years as in the study presented here (2013-2015). They seem to have somewhat lower RMSDs (4.1...5.4 cm) as compared to the results presented here (4.9...6.7 cm), although their results are independent of OIB measurements, while the results here are not. Why are these results not compared here? Is there any advantage of using the method presented here as compared to the one used in Guerreiro et al. (2016)? This comparison and discussion is missing here!

I found it confusing that the authors first declare that radar altimetry penetrates through to the snow-ice-interface, while laser altimetry does not (p. 2). AltiKa is presented as a radar altimeter (thus suggesting that it penetrates through the snow), but it is later compared with OIB's ATM laser freeboard (section 3.5). From what I understood, Guerreiro et al. (2016) conclude that the radar signal from AltiKa does not penetrate the snow, while Armitage and Ridout (2015) concluded that the AltiKa signal is scattered from roughly the midpoint of the snow layer. This discrepancy is not even mentioned here. What do your results suggest? Please comment/discuss/specify.

Another issue is that I think the structure of the paper could be improved:

a) In an "Introduction" I would mainly expect to read about the importance of the presented study, how it fits into the context of already existing studies and what is the new contribution of the presented study. Instead, we here get a general introduction on the importance of snow (ok) and we are presented the equations used to convert ice/ snow freeboard to snow depth (more appropriate for the "Data and Methods" section?). This is followed by a chapter that lists existing snow depth products, where I would prefer to read more about the differences to the presented study and the implications these have instead of a list of methods.

b) The "Results" section contains a lot of what I would consider discussion (or speculation as some of the statements on p. 11 are not based on citations), while the "Discussion" section on p. 14, l. 20 starts with showing more results...

Otherwise, the manuscript is, in general, well written and I was able to follow the method.

Specific comments:

p. 1, l. 3: "*...can be applied to any coincident freeboard measurements*" -> to any coincident snow and ice freeboard measurements? (would be clearer)

p. 1, l. 19: "*...snow depth estimates could be usefully assimilated...*" -> "usefully" is a vague (and strange) expression here...

p. 1, l. 23-24: "*The implications ... is*" -> The implications ... are

p. 2, l. 4: Eq. (1) -> Is this formula from Beaven et al., 1995?

p. 2, l. 27-28: "*The granular nature of snow acts to scatter and dissipate microwave energy radiating from the Earth's surface, reducing the surface brightness temperature.*" -> This statement is only true for part of the frequency spectrum of microwaves! Not true for low microwave frequencies.

p. 2, l. 30: "*for a given frequency*" -> Too vague, I'd prefer to see the frequency (range) that you mean here.

p. 3, l. 30-31: "*AltiKa was designated with a maximum penetration depth of 0, i.e. no penetration, and CS-2 a maximum penetration of 1, i.e. full snow penetration...*" -> What does this mean? Is it possible to retrieve snow depth using this method? Could you compare these with your method?

p. 4, l. 8-14: You write about the issues raised by different satellite footprint sizes, please also give the CS-2 footprint size here to make the comparison easier.

p. 4, l. 28: "*retrieves surface elevations up to 81.5∘*" -> a) Please add "latitude" (to avoid confusion with "geometrical elevations", which can also be given in degrees). b) I think this should be mentioned earlier in the manuscript because it constitutes a major limitation for polar applications of AltiKa.

p. 6: References for statements in l. 10-15 ?

p. 6, l. 20: "*It*" -> it + "*this criteria*" -> this criterion

p. 6, l. 21: Is "*snagging*" a word generally used for this? (just asking)

p. 6, l. 22: "*To overcome these problems,...* " -> refers to which problems? the off-nadir ranging of leads or also roughness?

p. 6, l.26: "*... we instead adopt an approach...*" -> Did you come up with this approach? Or did you take up an existing approach? (If yes, which one?/Reference?)

p. 6, l. 30-31: "*the appeal of this methodology is its applicability to any freeboard data sets*" ->

Why would this (i.e. applying to any freeboard data sets) not be possible for the method described in Guerreiro et al. (2016), for example? Wouldn't both have to be re-evaluated for their performance with different freeboard measurements anyways?

p. 6, l. 31: "*By calibrating satellite freeboards with an independent data set, biases are systematically corrected for*" -> I think this statement is too "optimistic"/general. Whether or not biases are systematically corrected for depends to a large extent on the quality, accuracy, and temporal + spatial resolution of the independent data. Not to mention that the bias is not the only thing to worry about...

p. 7, l. 7: "*snow depth, retrieved with the Kansas Snow Radar to within 5 cm accuracy*" -> Here (and also already in the introduction) it should be mentioned that different snow depth retrieval algorithms give very different snow depths! (Kwok et al., 2017)

p. 12, l. 5: Asterisk too high?

p. 14, l. 22: "*Spring*" -> spring

p. 16, l. 3-5: Did you use 2016 OIB data for calibration when comparing with the 2013, 2014 and 2015 OIB data? If not, why not?

p. 16, l. 14-15: remove parentheses around "*Kwok et al., 2017*"

p. 17, l. 1: "*onto a onto a*"

p. 17, 9-10: "*Snow depth agrees with expected spatial distribution and magnitude*" -> Compared to what? How do you know? Or do you mean just with regard to the statement that follows (on thicker snow over multi-year and thinner snow over first-year ice)?

Fig. 18, l. 8: "*...this evaluation does demonstrate the ability to up-scale OIB snow depths to the wider-Arctic*" -> Do you mean ability as in "we do not get nonphysical snow depth values" or how is the ability for this demonstrated here without comparing the snow depths to independent data?

Fig. 1 & 2: For the sake of completeness, it would be good to mention what the dashed grey line is (the zero line I guess).

Fig. 3: Why are the snow depths smoothed? Is there a physical reason for this? The spatial variability contains information too (about real variability or about the "consistency" of the method, for example), why not show this?

Fig. 6: It is very hard to see the OIB measurements on top of the snow depth map. Maybe zooming into the campaign area would be useful? L. 3 of caption: "*each grid cells*" -> each grid cell"

Fig 6 & 7: In none of the scatter plots there is snow depth values <0cm or >60cm, why would you show the data for a range of -20 to 80cm? This raises the question whether this was made to make the regression look "better"... and also creates unused white space that could be used for information instead.

---

## Author Comment (AC1) · 15 Aug 2018

k. Guerreiro (Referee)

kevin.guerreiro@legos.obs-mip.fr

Review for "Estimating snow depth over arctic sea ice from calibrated dual-frequency radar freeboards" by Lawrence et al.

General comments:

The study "Estimating snow depth over arctic sea ice from calibrated dual-frequency radar freeboards " by Lawrence et al. uses a combination of altimeters operating at different frequencies (Ku and Ka bands) and flying over the Arctic at the same period (2013-2016) in order to estimate snow depth at the top of sea ice.

Based on previous studies, the authors consider that the main return of the Ka-band radar signal arises from an upper part of the snowpack and that the main return of the Ku-band radar signal originates from a lower part of the snow pack. Using this difference of penetration depth into the snowpack, they estimate snow depth at the top of sea ice by calculating the difference of freeboard height between SARAL/AltiKa (Ka-band) and CryoSat-2 (Ku-band). Before processing the freeboard difference, the authors correct freeboard biases related to radar penetration/surface state. To correct these biases, they fit their Ka and Ku freeboard measurements using laser and radar measurements from the Operation Ice Bridge (OIB) 2013-2016 campaigns. To validate their "Dual-altimeter Snow Depth" estimates, the authors use "independent" snow depth measurements from the Operation Ice Bridge airborne campaigns. Further, they show that the methodology derived with CryoSat-2 and AltiKa can be reproduced using Envisat (Ku-band) and ICESat (Laser).

The paper focuses on a very relevant topic as snow depth is one of the most important sources of uncertainties when converting ice freeboard to ice thickness. Hence, measuring snow depth at pan-arctic scale with a good temporal resolution could strongly help to improve current sea ice thickness estimates. In addition, snow depth is a key thermodynamics parameter as it isolates sea ice from the cold atmosphere in winter and reflects an important amount of solar radiations in summer. Being able to measure snow depth at large scale could therefore truly help to improve our understanding of sea ice growth and melt processes.

In my opinion, the approach of using a combination of altimeter measurements to estimate snow depth deserves publication. However, I have some major remarks that must be addressed before the paper can be published:

1) While the authors considers that the Ka and Ku radar signals do not penetrate identically into the snowpack, it is not clearly stated where the main returns arise from. The authors quote Armitage and Ridout (2015) and Guerreiro et al. (2016), which draw different conclusions, but they don't clearly give their thoughts. This precision is crucial as one needs to know if the freeboard fit they perform with OIB is used to correct footprint effects or/and penetration effects.

*Author Response (AR)*: **The idea of correcting for both biases due to footprint effects / surface state and physical penetration at once is adopted in order to avoid quantifying the actual penetration of each satellite. Based on radiative transfer theory we assume in general that Ku will penetrate further into the snowpack than Ka but we make no assumptions about how far either is penetrating. The final section of the introduction (lines 3-15 page 4) has been re-written to clarify this.**

The authors say "Freeboard estimates from CryoSat-2 (Ku-band) and AltiKa (Ka-band) are calibrated against data from NASA's Operation IceBridge (OIB) to align AltiKa to the snow surface and CryoSat-2 to the ice snow interface". Considering this sentence, I assume that they consider (as in Armitage and Ridout (2015)) that Ku does not fully penetrate the snowpack while Ka does penetrate it a little bit, right?

*AR*: **We assume that Ku penetrates further into the snow pack than Ka, and therefore choose to correct CryoSat-2 to the ice/snow interface and Ka to the snow surface. We make no assumptions about how far into the snow each is penetrating since we do not think the effects of snow penetration can be separated from biases due to footprint size and surface effects.**

If yes, this raises an important question that should be answered more clearly: why the penetration of the Ka and Ku-bands would change from one area to another (Figure 1 and 2 show that the corrections are not constant)?

*AR*: **Figures 1 and 2 do not demonstrate that the snow penetration varies but rather that the combined effects of snow penetration and footprint/surface biases vary from one area to another. Again, the idea with this methodology is that nowhere do we separate the influence of the two. Ideas for why the combined effects of snow penetration and footprint/surface biases ($\Delta f$) varies from one area to another are given in the analysis of figures 1 and 2 (lines 18-25 page 8 and line 32 page 8 onwards).**

Also, this assumption seems to not take into account the results shown in Kurtz et al. (2014) and Guerreiro et al. (2017), why?

*AR*: **Kurtz et al. (2014) and Guerreiro et al. (2017) demonstrate the importance for elevation retrieval of surface properties and footprint size respectively. Due to these findings, we do not attribute the deviation of retrieved freeboards from snow and ice freeboard respectively as being due to snow penetration differences but a combination of penetration differences and biases due to sampling area. A reference to the findings of Kurtz et al. (2014), not included in the original manuscript, is now included in section 2.3 (lines 16 – 19 page 6), while references to the findings of Guerreiro et al. (2017) are given in lines 27-31 page 3, lines 6-9 page 4, and lines 12-14 page 5.**

2) The "error calculation section" (4.2) deserves some improvements. First of all, the authors calculate the uncertainty from an error propagation using a quadratic formula with variables that are clearly not independent. The variables covariance should be taken into account to avoid this issue.

*AR*: **We agree that covariance is required and have updated the formula and discussion in this section to account for this.**

3) Also, they consider that AltiKa and CryoSat-2 have a similar standard deviation on sea surface estimate and they come to the conclusion that, since AltiKa coverage is better than CryoSat-2, AltiKa freeboard error is smaller than that of CryoSat-2. In my opinion, this cannot be true even with the better coverage of AltiKa in the studied region. To derive appropriate

errors, the authors should calculate the standard deviation on sea surface for each satellite mission before injecting it in equation (6).

*AR*: **Following your suggestion, we now calculate the error on the sea-level interpolation for AltiKa and CryoSat-2 independently. The methodology for this is outlined in section 3.2.**

Finally, I am not sure what the authors mean by "correlation coefficient" in section 5.1. According to the values in Table 2, I am guessing that they calculate the fit regression line slope. I think it would be preferable to provide a Pearson coefficient R, which is a more common parameter.

*AR*: **We have replaced the correlation coefficient with the Pearson coefficient as suggested.**

4) I acknowledge that contemporary large-scale snow depth measurements are extremely rare and that using the same dataset for calibration and validation is one of the only existing options. Having said that, I would suggest to modify the plan of section 5.1 by not considering the year 2016 as a particular one (Figure 6). At the end of the day, Figure 6 (2016) and Figure 7 (2013-2014-2015) are almost identical: you remove observation from the considered year to evaluate your DuST snow depth. Thus, it does not require two figures nor two comments/conclusions.

*AR*: **We have combined evaluations for each year into a single figure and conclusion section as suggested.**

Minor comments:

Page 1 L 4-6: "Freeboard estimates . . . ice/snow interface". Does it mean that Ka/Ku don't stop at the air-snow/snow ice interface?

*AR*: **We make no assumption about where Ka/Ku penetrate to, only that Ku penetrates further than Ka and therefore we 'raise' Ka to the snow surface and 'push' Ku to the ice/snow interface. We feel this is now adequately explained in the introduction and does not require a clarification in the abstract.**

L 23: As you mention Envisat above, you should also quote Giles et al. (2008).

*AR*: **We have included Giles et al. (2008)**

Page 2: L15-18: This is arguable. For LRM altimeters, the uncertainty related to free-board height is at least as large as the one related to snow depth.

**AR: We no longer reference the results of Giles et al. (2007) in this section and now state that "For both the radar and laser case, snow depth is one of the dominant sources of sea ice thickness uncertainty". Please refer to page 1, lines 21 onwards.**

Page 3: L 31: To be more precise, Guerreiro et al. (2017) suggest that the Ka-band signal stops within the first few centimetres and that the Ku-band signal can stop before the snow-ice interface in case of large snow grains.

***AR***: **We have added this clarification (lines 9 to 18, page 3)**

L33: This is not exact: The first study that showed AltiKa freeboard measurements was the one by Maheshwari et al. (2015).

***AR***: **We have removed this claim.**

Page 4:

L13: Here and elsewhere, can you mention which footprint you talk about (beam-limited or pulse-limited).

***AR***: **We have specified which footprint we are referring to in the manuscript.**

L15: So here, you choose to follow the conclusion given in Armitage and Ridout (2015), which is that the Ka and Ku signals stop within the snow pack, right? If yes, could you state it more clearly? Also, considering the literature you quoted (or not) (Kurtz et al., 2014; Maheshwari et al., 2015; Guerreiro et al., 2016; Schwegman et al., 2015), could you please explain this choice. This is indeed a major point as your entire study is based on this assumption.

***AR***: **In response to your major criticism number (1), we feel we have now addressed this point. Lines 3 onwards, page 4, have been re-written accordingly.**

L29: Not exact: see my previous comments.

***AR***: **We have removed this claim.**

Page 6:

L3: In Armitage and Ridout (2015), I believe that the authors follow another condition related to the Leading Edge Width (see supplementary material). Could you check on that please?

***AR***: **In the AltiKa processing, the Leading Edge Width (LEW) is a criterion for identifying 'valid' waveforms but it applies equally to leads and floes: both must have a LEW less than 2 range bins else they are discarded (Armitage and Ridout, 2015, supplementary). LEW therefore is not used to discriminate leads from floes, which is the focus of our discussion in this section. Having said this, the backscatter coefficient Sigma0 is used to identify leads for AltiKa, and for CS-2 the Stack Standard Deviation (SSD) is used to differentiate leads from floes. Details of this have been added (page 5 lines 21 - 23)**

L26-31: To me, this way to proceed raises an important question: As you mention it above, the altimeter range can be biased by waveform hooking due to the proximity of specular reflections. Thus, if you calibrate your freeboards in a particular region (the one overflown by OIB for example), the calibration will likely depend on the density of Off-Nadir reflections found in this region. Consequently, the derived calibration might not work in regions where the density of Off-Nadir reflections is different. To check if your calibration is region-dependent or not, a simple test can be operated: you can plot the residuals of Figures 1 and 2 on a map and check if you observe regional patterns or not. This figure could be provided in the supplementary material.

**AR**: Thank you for this suggestion; this is something we originally considered but did not include since there was no evident regional dependence to the linear regression residuals. We have included the plots below for reference. The calibrations themselves, i.e. the extent to which satellite freeboard deviates from the snow or ice freeboard is of course region dependent and this is why we choose pulse peakiness as a means to characterize the surface. Our methodology assumes that surface properties including density of leads are sufficiently accounted for with the pulse peakiness criteria (low peakiness regions correspond to thicker multi-year ice where less leads are present; conversely we would expect highly peaky regions to be lead-dense) to extend the calibration beyond the region sampled by IceBridge.

AK delta freeboard calibration residuals map

[Figure]

Delta freeboard residual (cm)

CS2 delta freeboard calibration residuals map

[Figure]

Delta freeboard residual (cm)

Page 7:

L16-19: How do you evaluate the spatial and temporal resolution?

*AR*: **The spatial and temporal resolution that give the most number of grid cells with a minimum of 50 OIB and satellite points in each. This has been clarified in the manuscript (page 8 lines 7-8).**

L28-30: Could you give the correlation coefficient (Pearson's)?

*AR*: **This is now provided.**

Page 9:

L2-4: As you consider that the bias you fit is due to penetration effects, then yes, a fs > 0 would imply that the Ku-band signal penetrates through sea ice. However, if one considers that this bias is also due to surface properties (roughness for example), positive fs values would simply suggest that the empirical retracking you use is not adapted to sea ice surfaces. This was clearly demonstrated in the study by Kurtz et al. (2014). Could you provide with a more detailed comment by integrating this other aspect?

*AR:* **This is a good point, thank you for this suggestion. The results of Kurtz et al (2014) are now discussed in section 2.3 (lines 16 – 19 page 6). Lines 1-4 page 9 have been updated to include this consideration.**

Page 11:

L-15-16: As you do not clearly mention why you need to calibrate AltiKa and CryoSat-2 freeboards (penetration depth? surface properties?, . . .), this conclusion is hard to understand. Why would your calibration be different from on region to another? Because of snow properties? Lead density? Surface diffusivity? You need to give more details in order to provide a more convincing conclusion.

*AR: This sentence has been removed and this is now discussed on page 13, lines 8-12. We hope that the clarifications made previously (lines 3-15 page 4) will now make this discussion coherent.*

L29-31: Same remark as above.

Page 12:

L3: Shouldn't the title be "Uncertainty calculation"? An error should be relative to a "truth measurement". . .

*AR:* **Section title changed accordingly.**

Eq 6: As fAk and fAk are clearly not independent (see Figure 1), you must take into account their covariance to calculate the uncertainty.

*AR:* **We now consider variable covariance in our uncertainty calculation.**

Page 14:

L1-2: Why do you apply the same standard variation value as for CryoSat-2 (4 cm)? As far as I know AltiKa sea level standard deviation is much larger than that of CryoSat-2. I would recommend to re-calculate a standard deviation for the two datasets here in order to make sure you have the right values.

*AR:* **We now calculate CryoSat-2 and AltiKa freeboard uncertainty independently (section 4.2)**

Page 15: Figure 6: I am quite surprised about the r value you provide (0.73) considering the figure you show. How do you calculate this coefficient? It seems to me that you provide with the fit regression line slope. Am I right? If yes, I think it would be preferable to provide a Pearson coefficient R, which is more common parameter.

*AR:* **This is now provided.**

Page 16:

Table 2: Same remark as above.

Section 5.1: I don't understand why you consider 2016 as a particular year. As suggested by Table 2, the comparison for 2016 is almost identical as for the other years (except that you don't use 2016 to calibrate your DuST snow depth for the 2013-2015 periods). In my opinion you should not make any distinction between 2013-2015 and 2016 and re-write this section as such.

*AR:* **We have combined evaluations for each year into a single figure and conclusion section as suggested.**

L22: There is no link between the footprint size and the bandwidth. Also, you can have a similar footprint with 2 different frequencies depending on the antenna size.

*AR:* **Lines 16-17 page 18 updated accordingly.**

Page 17:
L1: "Onto a" is written twice

*AR:* **corrected**

L3-5: This description should be moved into the figure caption.

*AR:* **corrected**

L10: What does 30+ mean? Can you provide with a range of values instead?

*AR:* **corrected**

Page 18:

L19: Considering that you did not use validation data to validate your results, I would not use "demonstrated" here...

*AR:* **We have changed this to 'tested'**

---

## Author Comment (AC2) · 15 Aug 2018

**Author Responses (ARs) to each reviewer comment are in Bold.**

The paper "Estimating snow depth over Arctic sea ice from calibrated dual-frequency radar freeboards" by Lawrence et al. deals with estimating snow depth by combining satellite-based measurements of snow and ice freeboard. The method requires prior calibration with independent freeboard measurements. Here, CryoSat-2 and AltiKa satellite freeboard measurements are calibrated with airborne Operation IceBridge (OIB) measurements.

The latter raises one of my main concerns: The method, as presented here, relies on having reliable independent freeboard data, which at the moment is only provided by OIB data. However, there is disagreement within the science community on how to interpret the OIB radar measurements, i.e. different retrieval algorithms differ in the way air-snow and snow-ice interfaces are detected and localized. A recent paper by Kwok et al. (2017) showed that this caused OIB snow depths as retrieved from different groups to differ on average by up to 7 cm (for first-year ice) and 12 cm (for multi-year ice) for the 2013-2015 data (see Fig. 7 & 8 in Kwok et al., 2017), which is used in the paper presented here. The variability of snow depths is also quite different (so it is not just a constant bias between the different products). Though this problem is briefly mentioned in the paper presented here, this is only done rather late (on p. 14, l. 14), which does not represent how severe this issue is for the proposed method of retrieving snow depth. I think that this should be mentioned and discussed far earlier and with more emphasis because it has major implications on the usability and accuracy of the proposed retrieval method! Ideally, the authors would perform their comparison not only for OIB quicklook data, but also for (at least) one of the other OIB-based freeboard retrieval data sets to estimate how much this can influence the results.

**Author Response (AR): You have raised an important point which was not addressed adequately in the first submission of the paper. In the latest draft, we discuss the results of the Kwok et al. (2017) inter-comparison paper early on in section 2.4 (page 7 lines 15-23) when first introducing Operation IceBridge (OIB), and acknowledge that the variability of OIB Snow Radar data from different groups presents a limitation to our methodology. Following the results of the Kwok et al. (2017) study we now employ instead NASA JPL snow depths in our methodology. This data set shows best agreement with ERA-interim and the Warren climatology for the years 2013-2015 (Fig. 9, Kwok et al. (2017)). Regardless of the discrepancy between different OIB Snow Radar datasets, our methodology offers a means to extrapolate the OIB snow depth dataset (whichever is chosen or deemed "best") to the wider Arctic, both spatially and temporally. Our DuST product would benefit from the development of a next-generation Snow Radar data set, combining the best qualities of each existing processing algorithm, as is advocated by Kwok et al. (2017). This is discussed in section 3.2 (page 13 line 14 to page 14 line 6). The use of an optimised OIB Snow Radar dataset could improve our snow depth estimates but would not alter the methodology, which we feel therefore deserves publication.**

A further concern is that the study of Guerreiro et al. (2016) also uses CryoSat-2 and AltiKa freeboard measurements to retrieve snow depth. Instead of calibrating these Ku and Ka-band measurements with independent data (as done here), they theoretically analyze the penetration depths of both radar altimeters in snow and use snow density estimates to modify the Ku-band radar signal's velocity through the snow. In their study, they compare their retrieved snow depths with OIB snow depths for the same years as in the study presented here (2013-2015). They seem to have somewhat lower RMSDs (4.1...5.4 cm) as compared to the results presented here (4.9...6.7 cm), although their results are independent of OIB measurements, while the results here are not. Why are these results not compared here? Is there any advantage of using the

method presented here as compared to the one used in Guerreiro et al. (2016)? This comparison and discussion is missing here!

**AR: We now include a discussion of the Guerreiro et al. (2016) approach and outline why our methodology is different and has its own advantages (page 7 lines 1-13). We have aimed in this paper to outline a methodology that can be applied in future when AltiKa is no longer operational, and demonstrate the methodology applied to Envisat and ICESat in order to show that it could also be applied to CS-2 and ICESat-2 when ICESat-2 is launched. The method of Guerreiro et al. (2016) relies on the ability to reprocess CS-2 data to produce pseudo LRM CS-2 data in order to achieve a footprint similar to AltiKa. By doing so, the authors can then assume that the remaining elevation difference found between AltiKa and pLRM CS-2 is the result of snow penetration difference alone and thus snow depth can be found as the difference between the two. This methodology could not be applied to, for example, CS-2 and ICESat-2 because neither dataset can be processed such as to make the effective footprints the same.**

I found it confusing that the authors first declare that radar altimetry penetrates through to the snow-ice-interface, while laser altimetry does not (p. 2). AltiKa is presented as a radar altimeter (thus suggesting that it penetrates through the snow), but it is later compared with OIB's ATM laser freeboard (section 3.5). From what I understood, Guerreiro et al. (2016) conclude that the radar signal from AltiKa does not penetrate the snow, while Armitage and Ridout (2015) concluded that the AltiKa signal is scattered from roughly the midpoint of the snow layer. This discrepancy is not even mentioned here. What do your results suggest? Please comment/discuss/specify.

**AR: Please re-refer to the introduction since it has been restructured in line with your subsequent criticism. Following an overview of the Guerreiro et al. (2016) and Armitage and Ridout (2015) studies, we have added a paragraph (page 4, lines 3 - 22) to clarify our position on AltiKa and CS2 snow penetration.**

Another issue is that I think the structure of the paper could be improved:

a) In an "Introduction" I would mainly expect to read about the importance of the presented study, how it fits into the context of already existing studies and what is the new contribution of the presented study. Instead, we here get a general introduction on the importance of snow (ok) and we are presented the equations used to convert ice/ snow freeboard to snow depth (more appropriate for the "Data and Methods" section?). This is followed by a chapter that lists existing snow depth products, where I would prefer to read more about the differences to the presented study and the implications these have instead of a list of methods.

**AR: On your advice, we have changed the structure of the paper. The introduction no longer contains any equations but is rather an overview of: the importance of snow, the current methods to retrieve it, their limitations, and our proposed methodology and a justification for its necessity.**

b) The "Results" section contains a lot of what I would consider discussion (or speculation as some of the statements on p. 11 are not based on citations), while the "Discussion" section on p. 14, l. 20 starts with showing more results...

**AR: We have now combined Results and discussion into one section.**

Otherwise, the manuscript is, in general, well written and I was able to follow the method.

Specific comments:

p. 1, l. 3: "*...can be applied to any coincident freeboard measurements*" -> to any coincident snow and ice freeboard measurements? (would be clearer)

**AR: The methodology can be applied to any coincident satellite freeboards they do not have to measure the snow and ice freeboards (indeed we suggest that AltiKa and CS-2 do not measure snow and ice freeboard but rather that we do not know where in the snowpack their signal originates).**

p. 1, l. 19: "*...snow depth estimates could be usefully assimilated...*" -> "usefully" is a vague (and strange) expression here...

**AR: This has been removed. See page 1 lines 18-20.**

p. 1, l. 23-24: "*The implications ... is*" -> The implications ... are

**AR: removed**

p. 2, l. 4: Eq. (1) -> Is this formula from Beaven et al., 1995?

**AR: This section of the introduction including this formula has been removed**

p. 2, l. 27-28: " *The granular nature of snow acts to scatter and dissipate microwave energy radiating from the Earth's surface, reducing the surface brightness temperature.*" -> This statement is only true for part of the frequency spectrum of microwaves! Not true for low microwave frequencies.

**AR: This statement has been removed. See page 2 line 19 onwards.**

p. 2, l. 30: "*for a given frequency*" -> Too vague, I'd prefer to see the frequency (range) that you mean here.

**AR: This section has been condensed. See page 2 line 19 onwards.**

p. 3, l. 30-31: "*AltiKa was designated with a maximum penetration depth of 0, i.e. no penetration, and CS-2 a maximum penetration of 1, i.e. full snow penetration...*" -> What does this mean? Is it possible to retrieve snow depth using this method? Could you compare these with your method?

**AR: This statement has been clarified (see page 3, lines 14-19). We now discuss their methodology and its limitations further on page 6 (lines 34 onwards).**

p. 4, l. 8-14: You write about the issues raised by different satellite footprint sizes, please also give the CS-2 footprint size here to make the comparison easier.

**AR: This has been included (page 3, line 30)**

p. 4, l. 28: "*retrieves surface elevations up to 81.5°*" -> a) Please add "latitude" (to avoid confusion with "geometrical elevations", which can also be given in degrees). b) I think this should be mentioned earlier in the manuscript because it constitutes a major limitation for polar applications of AltiKa.

**AR: "latitude" has been included (page 4, line 27). This is now also mentioned earlier on page 3, (line 11).**

p. 6: References for statements in l. 10-15 ?

**AR: Added (page 6, line 16)**

p. 6, l. 20: "*It*" -> it + "*this criteria*" -> this criterion

**AR: Corrected (page 6, line 23)**

p. 6, l. 21: Is "*snagging*" a word generally used for this? (just asking)

**AR: Yes. Not sure when the term was coined but it appears as early as 1992 in "F. M. Fetterer, M. R. Drinkwater, K. C. Jezek, S. W. C. Laxon, R. G. Onstott, and L.M. H. Ulander, "Sea Ice Altimetry," in Microwave Remote Sensing of Sea Ice. Washington, DC, USA: AGU, 1992, ch. 7"**

p. 6, l. 22: "*To overcome these problems,...* " -> refers to which problems? the off-nadir ranging of leads or also roughness?

**AR: This has been changed to "To overcome the CS-2/AltiKa freeboard bias" (page 6, line 26)**

p. 6, l.26: "*... we instead adopt an approach...*" -> Did you come up with this approach? Or did you take up an existing approach? (If yes, which one?/Reference?)

**AR: We came up with this approach. We have changed this to "we here introduce an approach" accordingly (page 6 line 31)**

p. 6, l. 30-31: "*the appeal of this methodology is its applicability to any freeboard data sets*"

Why would this (i.e. applying to any freeboard data sets) not be possible for the method described in Guerreiro et al. (2016), for example? Wouldn't both have to be re-evaluated for their performance with different freeboard measurements anyways?

**AR: The method of Guerreiro et al. (2016) relies on the ability to reprocess CS-2 data to produce pseudo LRM CS-2 data in order to achieve a footprint similar to AltiKa. By doing so, the authors then make the assumption that the remaining elevation difference found between AltiKa and pLRM CS-2 is the result of snow penetration difference alone and thus snow depth can be found as the difference between the two. This methodology could not be applied to, for example, CS-2 and ICESat-2 because neither dataset can be processed such as to make the effective footprints the same. This is explained in section 2.3, page 7 line 5 onwards.**

p. 6, l. 31: "*By calibrating satellite freeboards with an independent data set, biases are systematically corrected for*" -> I think this statement is too "optimistic"/general. Whether or not biases are systematically corrected for depends to a large extent on the quality, accuracy, and temporal + spatial resolution of the independent data. Not to mention that the bias is not the only thing to worry about...

**AR: This sentence has been removed**

p. 7, l. 7: "*snow depth, retrieved with the Kansas Snow Radar to within 5 cm accuracy* " -> Here (and also already in the introduction) it should be mentioned that different snow depth retrieval algorithms give very different snow depths! (Kwok et al., 2017)

**AR: This is now discussed in this section (see page 7 lines 15-23)**

p. 12, l. 5: Asterisk too high?

**AR: Asterisk has been removed. (Page 14 line 9)**

p. 14, l. 22: "*Spring*" -> spring

**AR: corrected**

p. 16, l. 3-5: Did you use 2016 OIB data for calibration when comparing with the 2013, 2014 and 2015 OIB data? If not, why not?

**AR: yes we did. This has been clarified (page 17, line 11)**

p. 16, l. 14-15: remove parentheses around "*Kwok et al., 2017*"

**AR: corrected**

p. 17, l. 1: "*onto a onto a*"

**AR: corrected**

p. 17, 9-10: "*Snow depth agrees with expected spatial distribution and magnitude*" -> Compared to what? How do you know? Or do you mean just with regard to the statement that follows (on thicker snow over multi-year and thinner snow over first-year ice)?

**AR: With regard to the statement that follows. This has been clarified (page 18, line 28 onwards)**

Fig. 18, l. 8: " *...this evaluation does demonstrate the ability to up-scale OIB snow depths to the wider-Arctic* " -> Do you mean ability as in "we do not get nonphysical snow depth values" or how is the ability for this demonstrated here without comparing the snow depths to independent data?

**AR: we mean that the scatter plots suggest the ability to use the calibration functions to predict OIB snow depths for an unsampled year and region. This has been clarified (page 19, line 13)**

Fig. 1 & 2: For the sake of completeness, it would be good to mention what the dashed grey line is (the zero line I guess).

**AR: included**

Fig. 3: Why are the snow depths smoothed? Is there a physical reason for this? The spatial variability contains information too (about real variability or about the "consistency" of the method, for example), why not show this?

**AR: Snow depth maps are now shown unsmoothed.**

Fig. 6: It is very hard to see the OIB measurements on top of the snow depth map. Maybe zooming into the campaign area would be useful? L. 3 of caption: "*each grid cells*" -> each grid cell"

**AR: Maps are no longer included and scatter plots for all years have been combined into a single plot.**

Fig 6 & 7: In none of the scatter plots there is snow depth values <0cm or >60cm, why would you show the data for a range of -20 to 80cm? This raises the question whether this was made to make the regression look "better"... and also creates unused white space that could be used for information instead.

**AR: Plots now scaled from 0-60 cm**

---

## Referee Report (RR1)

**Review for « Estimating snow depth over arctic sea ice from calibrated dual-frequency radar freeboards » by Lawrence et al.**

**General comments :**

      The authors have done lots of efforts to improve the quality of their manuscript, providing all the supplementary work asked after the first review. For this reason, I recommend the study "Estimating snow depth over arctic sea ice from calibrated dual-frequency radar freeboards" by Lawrence et al. for publication. Congratulations to the authors for this excellent study.

**Technical corrections :**

Page 1:

L22-23: ICESat does not provide sea ice freeboard but snow freeboard. Please, consider modifying this sentence.

Page 2:

L4 and elsewhere: "In situ" is a latin expression, there is no hyphen.

Page 4:

L12: "of the surface and compare". Please consider adding "roughness" or "properties".